# Inactivation of *ackA* and *pta* Genes Reduces GlpT Expression and Susceptibility to Fosfomycin in *Escherichia coli*

Hidetada Hirakawa,[a] Ayako Takita,[a] Yumika Sato,[a] Suguru Hiramoto,[b] Yusuke Hashimoto,[a] Noriyasu Ohshima,[c] Yoji A. Minamishima,[c] Masami Murakami,[b] Haruyoshi Tomita[a,d]

[a]Department of Bacteriology, Gunma University Graduate School of Medicine, Gunma, Japan
[b]Department of Clinical Laboratory Medicine, Gunma University Graduate School of Medicine, Gunma, Japan
[c]Department of Biochemistry, Gunma University Graduate School of Medicine, Gunma, Japan
[d]Laboratory of Bacterial Drug Resistance, Gunma University Graduate School of Medicine, Gunma, Japan

**ABSTRACT** Fosfomycin is used to treat a variety of bacterial infections, including urinary tract infections caused by *Escherichia coli*. In recent years, quinolone-resistant and extended-spectrum $\beta$-lactamase (ESBL)-producing bacteria have been increasing. Because fosfomycin is effective against many of these drug-resistant bacteria, the clinical importance of fosfomycin is increasing. Against this background, information on the mechanisms of resistance and the antimicrobial activity of this drug is desired to enhance the usefulness of fosfomycin therapy. In this study, we aimed to explore novel factors affecting the antimicrobial activity of fosfomycin. Here, we found that *ackA* and *pta* contribute to fosfomycin activity against *E. coli*. *ackA* and *pta* mutant *E. coli* had reduced fosfomycin uptake capacity and became less sensitive to this drug. In addition, *ackA* and *pta* mutants had decreased expression of *glpT* that encodes one of the fosfomycin transporters. Expression of *glpT* is enhanced by a nucleoid-associated protein, Fis. We found that mutations in *ackA* and *pta* also caused a decrease in *fis* expression. Thus, we interpret the decrease in *glpT* expression in *ackA* and *pta* defective strains to be due to a decrease in Fis levels in these mutants. Furthermore, *ackA* and *pta* are conserved in multidrug-resistant *E. coli* isolated from patients with pyelonephritis and enterohemorrhagic *E. coli*, and deletion of *ackA* and *pta* from these strains resulted in decreased susceptibility to fosfomycin. These results suggest that *ackA* and *pta* in *E. coli* contribute to fosfomycin activity and that mutation of these genes may pose a risk of reducing the effect of fosfomycin.

**IMPORTANCE** The spread of drug-resistant bacteria is a major threat in the field of medicine. Although fosfomycin is an old type of antimicrobial agent, it has recently come back into the limelight because of its effectiveness against many drug-resistant bacteria, including quinolone-resistant and ESBL-producing bacteria. Since fosfomycin is taken up into the bacteria by GlpT and UhpT transporters, its antimicrobial activity fluctuates with changes in GlpT and UhpT function and expression. In this study, we found that inactivation of the *ackA* and *pta* genes responsible for the acetic acid metabolism system reduced GlpT expression and fosfomycin activity. In other words, this study shows a new genetic mutation that leads to fosfomycin resistance in bacteria. The results of this study will lead to further understanding of the mechanism of fosfomycin resistance and the creation of new ideas to enhance fosfomycin therapy.

**KEYWORDS** antimicrobial resistance, AMR, fosfomycin, urinary tract infection, UTI, bacterial pathogenesis, molecular genetics, infection control

Address correspondence to Hidetada Hirakawa, hirakawa@gunma-u.ac.jp.

The authors declare no conflict of interest.

Urinary tract infection (UTI) is one of the most common infectious diseases. Approximately 50% of women experience UTIs in their lifetimes (1, 2). *Escherichia coli* is a major pathogen that is estimated to cause over 80% of uncomplicated UTIs (3,

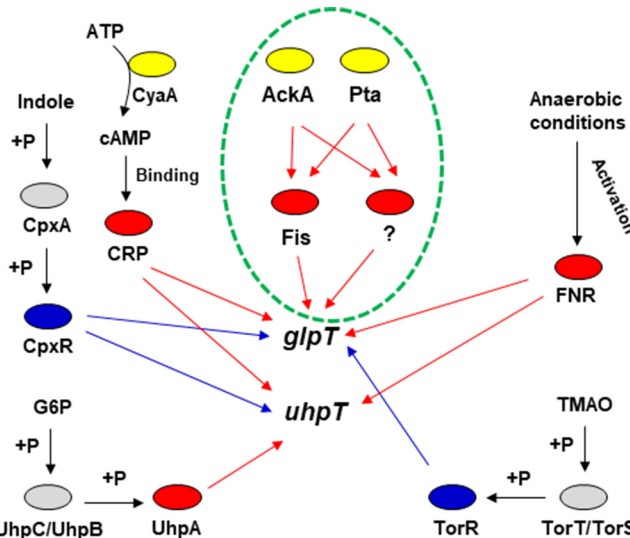

**FIG 1** Overview of the regulatory mechanism of *glpT* and *uhpT* expression and the mechanism of increased fosfomycin resistance by *ackA* and *pta* mutations. The red ovals are activators for expression of *glpT* and/or *uhpT*, while the blue ovals are repressors. The gray ovals indicate proteins that sense small molecules and phosphorylate downstream regulators. The ovals indicated in yellow are metabolic enzymes. Red and blue arrows indicate increased and decreased expression of target genes, respectively. The finding in this study is highlighted in a green dashed circle. P, phosphate group; TMAO, trimethylamine-*N*-oxide; G6P, glucose-6-phosphate.

4). Some classes of antimicrobial agents such as quinolones, $\beta$-lactams, and trimethoprim-sulfamethoxazole have been used for treatment of UTIs. However, increasing numbers of bacteria have acquired resistance to antimicrobials (5–7). Despite growing awareness of this issue, there is a shortage of new alternative agents.

Fosfomycin is used to treat *E. coli* infections, including UTIs. This drug is still effective against many *E. coli* strains that are resistant to commonly used antimicrobials such as quinolone- and trimethoprim-sulfamethoxazole-resistant strains and extended-spectrum $\beta$-lactamase (ESBL) producers because its structure is distinct from other known antimicrobials, hence there is no cross-resistance issue (8, 9). For this reason, fosfomycin has recently reattracted attention as a last resort agent to treat multidrug-resistant (MDR) *E. coli* strains.

Fosfomycin enters the cells via GlpT and UhpT, glycerol-3-phosphate and glucose-6-phosphate transporters, respectively, and then inhibits the UDP-*N*-acetylglucosamine-3-enolpyruvyltransferase (MurA) activity required for bacterial cell wall biosynthesis (10). Susceptibility to fosfomycin can be affected by several genetic modifications. Acquisition of the *fosA*, *fosB*, *fosX*, *fomA*, and *fomB* genes leads to inactivation of fosfomycin (11–14, 15). In addition to these exogenous mechanisms, functional mutations in the genes encoding GlpT, UhpT, and MurA on the chromosome alter susceptibility to the drug. In particular, functional deletion of GlpT and UhpT and substitution of the Cys115 residue in MurA, which forms a covalent bond with fosfomycin, cause reduced sensitivity to fosfomycin (16–18).

Fosfomycin sensitivity is also attenuated by decreased expression of GlpT and UhpT, which can occur by several mechanisms (Fig. 1). Fosfomycin sensitivity is also attenuated by decreased expression of GlpT and UhpT. CRP-cAMP (cyclic AMP receptor protein-cyclic AMP) and UhpA are positive regulators for *glpT* and *uhpT* genes. Genetic mutations in CRP, CyaA and PtsI (required for cAMP production), and UhpA reduced GlpT and UhpT expression and decreased fosfomycin uptake and sensitivity (19–21, 22). UhpA is phosphorylated via histidine kinase UhpB and membrane-bound protein UhpC, which induces the expression of *uhpT*. In a recent study, mutations that inactivate the genes for *uhpB* and *uhpC* were found to result in reduced expression of *uhpT*, with consequent resistance to fosfomycin (23).

In addition to the positive regulators, CpxR and TorR, response regulators composing two-component regulatory systems act as repressors for *glpT* and *uhpT* genes (Fig. 1).

These regulators are activated in the presence of indole and trimethylamine-*N*-oxide (TMAO), respectively, and then they repress GlpT and UhpT expression. This induces a decrease in the activity of fosfomycin (24, 25). Thus, regulatory proteins associated with *glpT* and *uhpT* expression may be potential risk factors that attenuate the efficacy of the fosfomycin therapy. On the other hand, the activity of fosfomycin is promoted by increased expression of GlpT and UhpT. The fumarate and nitrate reductase (FNR) protein was shown to promote the expression of *glpT* and *uhpT* under anaerobic conditions, resulting in increased susceptibility to fosfomycin in *E. coli* (26).

We aimed to obtain further insight into the bacterial genes that affect fosfomycin activity. This insight will aid us to predict a resistance occurrence and establish a strategy that enhances the utility of this drug. Previously, we constructed a transposon (Tn*5*) random insertion library in an uropathogenic *E. coli* (UPEC) strain (27). At the beginning of the study, we carried out genetic screening to identify genes that contribute to fosfomycin activity. We found two strains that had transposon inserts in the *ackA* and *pta* genes, respectively.

The *ackA* and *pta* genes are involved in acetyl-coenzyme A (acetyl-CoA) degradation into acetate and ATP production. The *ackA* gene encodes acetate kinase that catalyzes a reaction to produce acetate from acetyl-phosphate, while the *pta* gene product is phosphate acetyltransferase, an enzyme to produce acetyl-phosphate from acetyl-CoA (28). These genes are cotranscribed in an operon (29). In this study, we show that inactivation of the *ackA* and *pta* genes reduces *glpT* expression and fosfomycin uptake, thereby causing reduced sensitivity to fosfomycin. Thus, the *ackA* and *pta* genes contribute to fosfomycin activity.

## RESULTS

**Deletion of the *ackA* and *pta* genes reduces susceptibility to fosfomycin.** Previously, we performed a random mutagenesis using the Epicentre EZ-Tn5<R6Kγori/KAN-2>Tnp Transposome kit to search for genes involved in biofilm formation in the UPEC CFT073 strain (27). In the process, we obtained approximately 2,200 clones with transposon insertions. The CFT073 genome contains 5,400 genes. The number of essential genes for culture in LB medium is estimated to be ~460 (30). Our library of 2,200 clones represents ~44% of nonessential genes.

To search for genes that contribute to the antimicrobial activity of fosfomycin, MICs of fosfomycin were determined for those clones. We focused on clones with MICs that increased >4-fold as moderately or more resistant strains. First, we found clones highly resistant to fosfomycin with transposons inserted in *cyaA* and *glpT*, respectively. In addition to these clones, we found only two other clones that were moderately resistant to fosfomycin (4-fold higher MICs than the parent strain). They had transposons inserted inside the *ackA* and *pta* genes, respectively.

To verify this result, we constructed in-frame deletion mutants of the *ackA* and *pta* genes (designated CFT073ΔackA and CFT073Δpta). Similar to the original strains with transposon insertions, CFT073ΔackA and CFT073Δpta showed 4-fold higher MICs than the parent strain (Table 1). MICs of fosfomycin in CFT073ΔackA and CFT073Δpta were reduced to parental levels by introducing the heterologous *ackA* and *pta* expression plasmids, pTrc99KackA and pTrc99Kpta, respectively (Table 1). We also compared the survival of the mutant strains to that of the parent strain at 1 to 3 h postexposure at 6.25× the MIC of the parent strain. Consistent with the findings of the MIC experiments, the *ackA* and *pta* mutants showed approximately 30- to 100-fold higher survival rates than the parent strain (Fig. 2).

Anti-*E. coli* activity of fosfomycin increases under anaerobic conditions (26). The effect of *ackA* and *pta* gene deletion was also determined in anaerobic cultures. In anaerobic culture, the MIC value of the CFT073 parent strain for fosfomycin was 4-fold lower than in aerobic culture. The *ackA* and *pta* mutants still exhibited 4-fold higher MICs than the parent strain even when assayed under anaerobic conditions (Table 2).

We also examined the effects of *ackA* and *pta* deficiency on the activity of some other antimicrobial agents, including levofloxacin, amikacin, tetracycline, piperacillin, cefotaxime,

**TABLE 1** Fosfomycin MICs of UPEC CFT073 and its derivatives

| Strain | MIC (mg/L) |
| --- | --- |
| CFT073 | 4 |
| CFT073ΔackA | 16 |
| CFT073Δpta | 16 |
| CFT073pTrc99K | 4 |
| CFT073ΔackA/pTrc99K | 16 |
| CFT073ΔackA/pTrc99KackA | 4 |
| CFT073 Δpta/pTrc99K | 16 |
| CFT073 Δpta/pTrc99Kpta | 4 |
| CFT073ΔackAΔpta | 16 |
| CFT073ΔglpT | 128 |
| CFT073ΔackAΔglpT | 128 |
| CFT073ΔptaAΔglpT | 128 |
| CFT073ΔuhpT | 16 |
| CFT073ΔackAΔuhpT | 64 |
| CFT073ΔptaΔuhpT | 64 |
| CFT073ΔglpR | 2 |
| CFT073ΔackAΔglpR | 8 |
| CFT073ΔptaAΔglpR | 8 |
| CFT073ΔcpxAR | 4 |
| CFT073ΔackAΔcpxAR | 16 |
| CFT073ΔptaAΔcpxAR | 16 |
| CFT073 Δfis | 8 |
| CFT073/pTrc99Kfis | 2 |
| CFT073ΔackA/pTrc99Kfis | 2 |
| CFT073Δpta/pTrc99Kfis | 2 |
| CFT073ΔglpT/pTrc99K | 128 |
| CFT073ΔglpT/pTrc99Kfis | 128 |

aztreonam, meropenem, and chloramphenicol. Unlike fosfomycin, we found no difference in MIC values between the parent and mutant strains (see Table S1 in the supplemental material).

**The *ackA* and *pta* mutants have lower level of *glpT* expression and fosfomycin uptake.** Reduced fosfomycin sensitivity in the *ackA* and *pta* mutants may be attributed to decreased drug uptake. To test this hypothesis, intracellular levels of fosfomycin in the *ackA* and *pta* mutants was compared to those in the parent strain. The levels in the *ackA* and *pta* mutants were 7.7- and 47-fold lower than the parent strain, respectively ($18.3 \pm 5.2$ $\mu$g/$10^8$ cells for the *ackA* mutant, $3.0 \pm 0.5$ $\mu$g/$10^8$ cells for the *pta* mutant and $140.4 \pm 67.4$ $\mu$g/$10^8$ cells for the parent strain) (Fig. 3A). Transcript levels of *glpT* and *uhpT* encoding transporters for fosfomycin uptake were also measured in these strains by quantitative PCR (qPCR) analysis. The transcript levels of *glpT* in the *ackA* and *pta* mutants were 4.0- and 4.3-fold lower than the parent strain (Fig. 4A). In contrast to the *glpT* transcript, no significant difference in the *uhpT* transcript was observed between the parent strain and the mutants (Fig. 4A). MurA is a target for the drug. We found no significant difference in

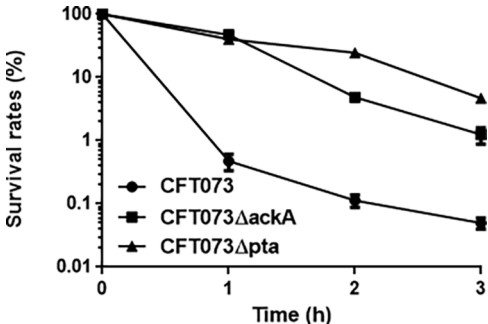

**FIG 2** Bacterial survival of CFT073 parent strain, the *ackA* mutant, and the *pta* mutant for fosfomycin. Bacteria were grown and then incubated with 25 mg/L fosfomycin. The survival rates at the indicated time points were described as the percent CFU for strains after incubation with fosfomycin relative to those before incubation with fosfomycin. Mean data are plotted, and error bars indicate the ranges.

**TABLE 2** Fosfomycin MICs in anaerobic cultures

| Strain | MIC (mg/L) |
|---|---|
| CFT073 | 1 |
| CFT073ΔackA | 4 |
| CFT073Δpta | 4 |
| CFT073ΔackAΔpta | 4 |

transcript levels of *murA* between the parent and mutant strains (Fig. 4A).

Fosfomycin is taken up separately via two different transporters, GlpT and UhpT. Based on the qPCR results, we hypothesized that the increased resistance to fosfomycin seen in the *ackA* and *pta* mutants is caused mostly by decreased *glpT* expression. If so, then we would expect to see no further increase in resistance to fosfomycin upon deletion of *ackA* and *pta* in the absence of *glpT*, while the absence of *uhpT* should not impact the effects of these mutations on fosfomycin sensitivity. The *ackA* and *pta* genes were deleted from the *glpT* mutant. The *glpT* mutant exhibited a 32-fold higher MIC to fosfomycin than the parent strain. However, no further increase of MIC values due to *ackA* and *pta* deletion was observed (Table 1). We also constructed *ackA* and *pta* deletion mutants in the *uhpT* mutant background. MIC of fosfomycin in the *uhpT* mutant was 4-fold higher than in the parent strain. In contrast to *glpT* mutant, deletion of *ackA* and *pta* from the *uhpT* mutant resulted in a further 4-fold increase in fosfomycin MICs (Table 1). Altogether, we conclude that increased resistance to fosfomycin in the *ackA* and *pta* mutants is due to decreased *glpT* expression.

**Reduced *glpT* expression in the *ackA* and *pta* mutants is associated with reduced expression of *fis* encoding a nucleoid-associated protein.** The expression of *glpT* is promoted by CRP-cAMP and FNR, while it is repressed by GlpR, CpxR, and TorR (21, 24–26, 31). However, we concluded that these regulators are not involved in the downregulation of *glpT* expression by *ackA* and *pta* deficiency for the following reasons. The CRP-cAMP activity is altered by intracellular cAMP levels (32). We found no significant difference in cAMP levels among the parental CFT073 strain, CFT073ΔackA and CFT073Δpta (Fig. 3B). Also, deletion of the *ackA* and *pta* genes from the *glpR* and *cpxR* mutants (CFT073ΔglpR and CFT073ΔcpxAR) still elevated the MIC of fosfomycin (Table 1). CpxR is activated by indole (33). Indole levels in these strains were essentially same (Fig. 3C). TorR primarily activates the transcription of *torC* (34). Transcript levels of *torC* between the parent and mutant strains were similar (Fig. 4B). No significant difference in transcript levels of *fnr* was also observed between the parent and mutant strains (Fig. 4B).

According to the database RegulonDB (35), nucleoid-associated proteins, Fis and IHF (IhfA/IhfB) and a DNA-binding transcriptional regulator, PlaR are presumed to regulate *glpT* expression. We measured the transcript levels of *fis*, *ihfA*, *ihfB*, and *plaR* and found that in the *ackA* and *pta* mutants, the transcript level of *fis* is approximately half that of the parent strain (Fig. 4B).

We hypothesized that Fis promotes *glpT* expression and that reduced expression of *fis* may be involved in reduced *glpT* expression. To test this, we introduced a reporter plasmid fused with the promoter of *glpT* and the promoterless *lacZ* gene into the parent strain and *fis* mutant and compared β-galactosidase activity depending on the expression level of LacZ. The *fis* mutant had significantly lower β-galactosidase activity than the parent strain (Fig. 5A). The MIC of the *fis* mutant to fosfomycin was determined and was 2-fold higher than that of the parent strain (Table 1). We also constructed an IPTG (isopropyl-β-D-thiogalactopyranoside)-inducible *fis* expression plasmid. Introduction of the parent CFT073 strain with pTrc99Kfis and overexpression of *fis* at 0.1 mM IPTG reduced the MIC to 2 mg/L, 2-fold lower than the parent strain (Table 1). Furthermore, when *fis* was overexpressed in *ackA* and *pta* mutants, the MIC to fosfomycin was 2 mg/L. On the other hand, when 0.1 mM IPTG was added to the *glpT* mutant carrying pTrc99Kfis, the MIC did not decrease. These observations suggest that the increased susceptibility to fosfomycin due to *fis* expression is dependent on *glpT* and that AckA and Pta act upstream of Fis in the regulation of *glpT*.

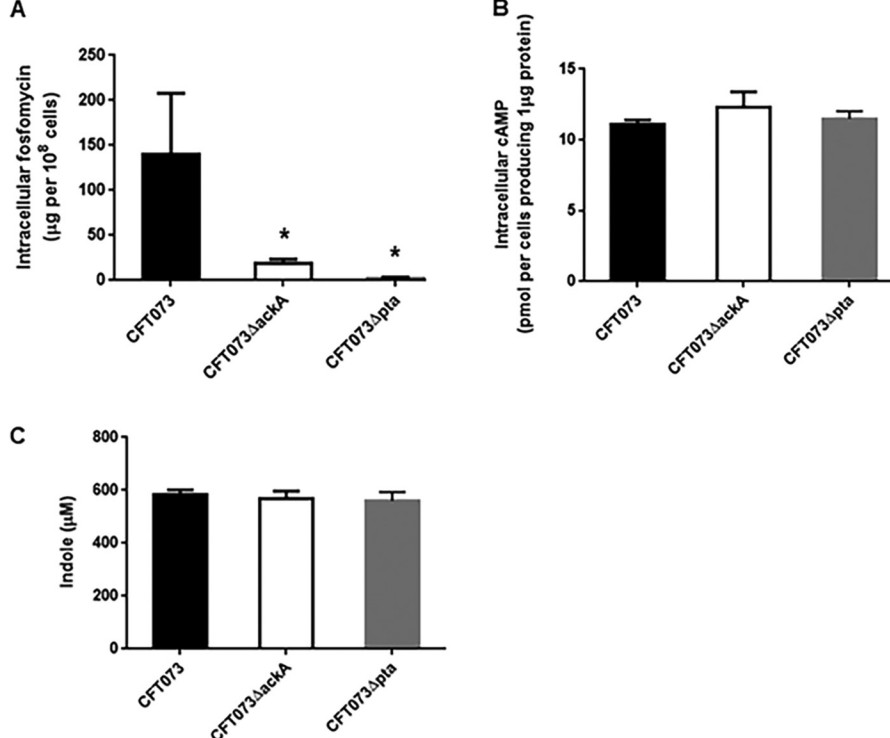

**FIG 3** (A to C) Assays for fosfomycin uptake (A), intracellular cAMP (B), and indole accumulation (C) in the CFT073 parent strain, the *ackA* mutant, and the *pta* mutant. (A) Intracellular accumulation of fosfomycin. Accumulation among these strains was described as amounts of fosfomycin ($\mu$g) in $10^8$ cells. (B) Intracellular concentration of cAMP. The intracellular concentration of cAMP was the amount of cAMP (represented as pmol) per cells producing 1 $\mu$g of protein. (C) Indole concentration in culture supernatant. Bacteria were aerobically grown in LB medium for 24 h. The indole concentration in culture supernatant was determined using Kovac's reagent as described in Materials and Methods. Mean data from three independent experiments are plotted; error bars indicate the standard deviations. *, $P < 0.05$ (relative to the value for CFT073).

We purified the Fis protein and examined its ability to bind to the upstream region of the *glpT* gene. We used a DNA probe consisting of ~330 bp containing the promoter of *glpT* and its upstream region and a negative control of ~300 bp of the upstream region of *Pseudomonas aeruginosa rhlR*. Gel shift assay results showed that the Fis protein showed a shift in the DNA band indicating binding to the *glpT* probe even at a concentration of only 0.25 pmol (Fig. 5B). In contrast, no shift in the *rhlR* DNA band was observed in the presence of 0.25 pmol of Fis (Fig. 5B). The shift could be seen at 1 pmol with the *rhlR* probe compared to 0.25 pmol with the *glpT* probe. These results suggest that Fis binds to the upstream region of *glpT*, increasing its promoter activity. Altogether, we conclude that *ackA* and *pta* mutations cause a decrease in *fis* expression and that the decrease in Fis level is involved in reduced *glpT* expression.

**Deletion of *ackA* and *pta* also results in increased resistance to fosfomycin in an ESBL-producing strain and in an enterohemorrhagic strain of *E. coli*.** To test whether genetic mutations in *ackA* and *pta* also reduce the antibacterial activity of fosfomycin in other clinically important *E. coli* strains, we selected two strains. One is a an UPEC ESBL producer that is resistant to quinolones, aminoglycosides and sulfamethoxazole-trimethoprim together with most of $\beta$-lactams (designated GU2019-E4). The other one is the well-characterized enterohemorrhagic *E. coli* strain (EHEC O157:H7 Sakai). To construct *ackA pta* double deletion mutants from these strains, respectively, the sequence of the *ackA-pta* operon for GU2019-E4 was determined because the complete genome sequence for this strain has not been available (see Fig. S1). We found orthologous *ackA* and *pta* genes in the GU2019-E4 chromosome. The *ackA pta* double deletion mutants in GU2019-E4 and EHEC O157:H7 Sakai showed 4-fold higher MICs of fosfomycin than their parental strains, respectively, as that observed in UPEC CFT073 (Table 3). The sequences of the *glpT* and *fis* open reading frames and the upstream regions of those genes for GU2019-E4 were

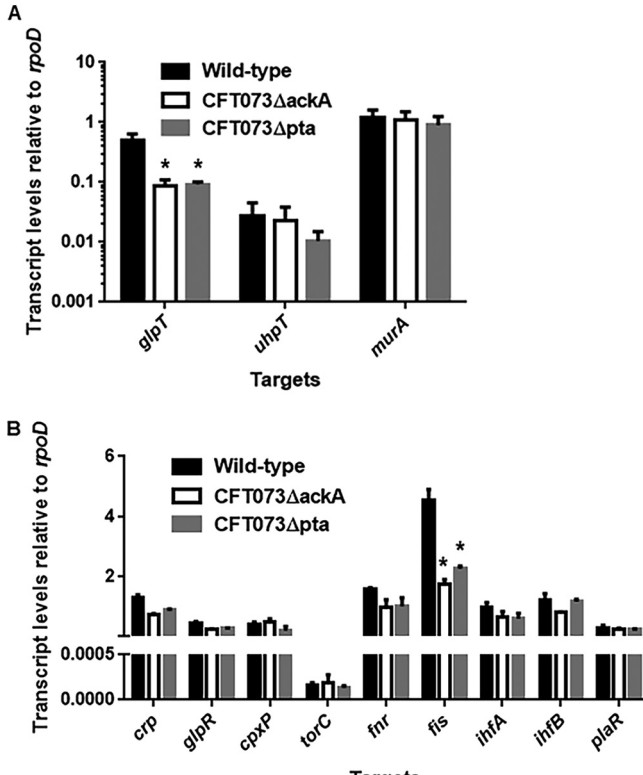

**FIG 4** (A and B) Transcript levels of *glpT*, *uhpT*, and *murA* genes (A) and regulatory genes for *glpT* (B) in the CFT073 parent strain, the *ackA* mutant, and the *pta* mutant. The transcript levels of target genes are presented as values relative to that of *rpoD* (the housekeeping gene). Data are plotted as the means from three independent experiments; error bars indicate the standard deviations. *, $P < 0.05$ (relative to the value for CFT073).

also determined and compared to those for CFT073 and EHEC O157:H7 Sakai strains (see Fig. S2 and S3). The sequences were >96% identical among the three strains. These observations suggest that inactivation of the *ackA* and *pta* genes reduces susceptibility to fosfomycin in the multidrug-resistant strain and EHEC, providing potential clinical relevance to mutations in these genes.

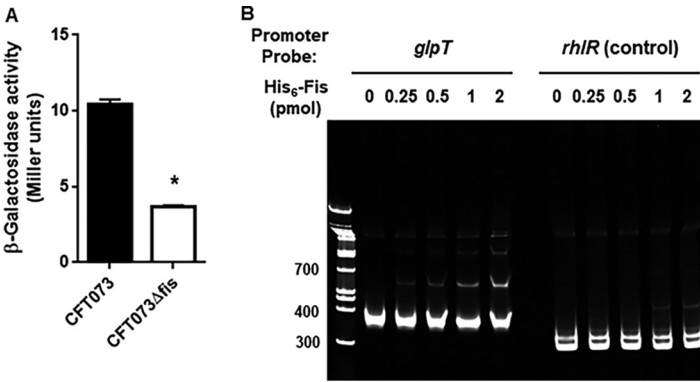

**FIG 5** (A and B) Promoter activity of *glpT* (A) and binding of Fis protein to the upstream region of the *glpT* gene (B). (A) *β*-Galactosidase activities from LacZ expression in CFT073 parent and the *fis* mutant correspond to *glpT* promoter activities are given in Miller units. Data plotted are as the means for three independent experiments; error bars indicate the standard deviations. *, $P < 0.05$ (relative to the value for CFT073). (B) Gel shift assay results showing the binding of Fis to the *glpT* upstream region. The Fis protein was added to reaction mixtures containing 0.3 pmol of DNA probe. Reaction mixtures were separated in polyacrylamide gels. Free and Fis-bound DNAs were visualized by Novel Green Plus staining under UV light (300 nm). The locations of DNA size standards (in base pairs) are indicated on the left.

**TABLE 3** Fosfomycin MICs of other clinical *E. coli* strains

| Strain | Fosfomycin MIC (mg/L) |
|---|---|
| GU2019-E4 (ESBL producer) | 8 |
| GU2019-E4ΔackAΔpta | 32 |
| O157 (EHEC) | 4 |
| O157ΔackAΔpta | 16 |

## DISCUSSION

It is known that chromosomal mutations can affect fosfomycin sensitivity. The transporters GlpT and UhpT are required for activity of fosfomycin because strains with mutations within the *glpT* and *uhpT* genes are less sensitive to the drug (19, 20). Bacterial sensitivity to fosfomycin also depends on expression level of *glpT* and *uhpT*. Expression of *glpT* and *uhpT* is promoted by CRP-cAMP and UhpA. Several clinical and laboratory studies demonstrated that mutations in *crp*, *cyaA*, and *uhpA* lead to decreased expression of *glpT* and *uhpT* and resistance to fosfomycin (19–21). The present study shows that AckA and Pta also contribute to expression of *glpT* and sensitivity to fosfomycin because deletion of the *ackA* and *pta* genes reduced expression of *glpT*, but not *uhpT*, and decreased sensitivity to the drug (Fig. 1). Therefore, the *ackA* and *pta* genes may be a source of fosfomycin resistance.

Recently, two comprehensive studies combining transposon mutagenesis with next-generation sequencing (NGS) analysis identified genes contributing to fosfomycin activity (36, 37). Interestingly, *ackA* and *pta* were not identified in these studies. One difference between our study and these is that in the other studies, mutants capable of surviving in the presence of fosfomycin were selected while in our study, mutants were obtained in the absence of fosfomycin. These differences could account for the differences in results obtained.

Expression of *glpT* is affected by the activity of several regulatory molecules such as CRP-cAMP, GlpR, FNR, CpxR, and TorR (21, 24–26, 31) (Fig. 1). According to results from fosfomycin sensitivity assays, small molecule (cAMP and indole) measurements, and qPCR gene expression analyses with our gene deletion strains, the decreased expression of *glpT* due to *ackA* and *pta* gene inactivation likely does not involve these regulators (Fig. 2, 3, and 4A) (Tables 1, 2, and 3). In addition to those regulators, Fis, IHF, and PlaR have been postulated to be involved in the regulation of *glpT* expression. In a series of experiments, including MIC assays, of *fis* deletion mutant, *fis* expression qPCR analyses, promoter assays, and gel shift assays, we showed that Fis increases *glpT* expression (Fig. 4B and 5) (Table 1). The qPCR results showed that the transcript level of *fis* is decreased in *ackA* and *pta* mutants, suggesting that the increased resistance to fosfomycin caused by *ackA* and *pta* mutations is related to the decreased expression of *fis* (Fig. 1). However, while the MIC of fosfomycin is increased 4-fold in *ackA* and *pta* deletion, the increase in MIC in *fis* deletion is only 2-fold. Thus, the increased resistance to fosfomycin caused by *ackA* and *pta* mutations cannot be explained by decreased Fis expression alone but may involve additional regulators.

The mechanism by which Fis levels are reduced by *ackA* and *pta* deletion is not yet known. Fis levels are known to be reduced in stationary phase during nutrient starvation (38, 39). Mutations in the *ackA* and *pta* genes affect a variety of metabolic activities, such as decreases in acetate production, glucose uptake, and the free CoA pool (40), which may cause decreased *fis* expression.

Acetyl-CoA generated in glycolysis is converted to acetyl-phosphate by Pta and subsequently degraded into acetate by AckA (29, 41). Therefore, inactivation of the *ackA* gene leads to accumulation of acetyl-phosphate (42). Acetyl-phosphate acts as a phosphate donor for some response regulators, and then their activities are promoted in the absence of AckA (43–45). Some studies showed that accumulation of acetyl-phosphate also causes increased lysine acetylation of protein, which alters protein activities (40, 46, 47). Our MIC data indicated that the *ackA pta* double mutant was less sensitive to fosfomycin compared to the wild-type parent strain to the same extent as

was the *ackA* single-deficient strain although the *ackA pta* double mutant cannot produce acetyl-phosphate (Table 1). Thus, decreased expression of *glpT* due to *ackA* and *pta* inactivation is associated with neither phosphorylation of particular response regulators nor protein lysine acetylation.

The *ackA* and *pta* gene products are required for mixed acid fermentation contributing to the ATP production under anaerobic conditions (48). The expression of *ackA* and *pta*, along with genes involved in acid resistance, increases in an FNR-dependent manner under anaerobic conditions (49, 50). For *E. coli*, oxygen-limited situations are very common in their habitat and during infection process. When they are in an enteric site, available oxygen may be limited by other microbial members competing with them. Uropathogenic strains, such as UPEC, encounter oxygen depletion within the bladder during UTI (51). For this reason, mutants lacking *ackA* and/or *pta* genes may be compromised in metabolic fitness. One study exhibited that the *ackA* and *pta* genes of UPEC are highly expressed in the bladder of UTI mice and the *ackA* and *pta* mutants are less virulent (52). Therefore, spontaneous mutations in *ackA* and/or *pta* may be disadvantageous *in vivo* for bacteria. However, it does not exclude the possibility that such mutants are generated by a fosfomycin pressure.

Mutations in the *ackA* and *pta* genes weaken the efficacy of fosfomycin, although their incidence is probably low due to the metabolic fitness burden. We do not yet know the clinical relevance of mutations in these genes. As more NGS data becomes available, it will be possible check if fosfomycin-resistant strains that have been isolated so far and those that will be isolated in the future have mutations in *ackA* and *pta*. We believe that this study will aid us to more precisely predict fosfomycin resistance and thereby help to keep fosfomycin treatment effective.

## MATERIALS AND METHODS

**Bacterial strains and culture conditions.** The bacterial strains and plasmids used in this study are listed in Table S2 in the supplemental material. The UPEC GU2019-E4 strain was originally isolated from the blood in a patient with pyelonephritis. This strain produces the ESBL enzyme and is resistant to piperacillin, cefotaxime, ceftazidime, aztreonam, levofloxacin, gentamicin, kanamycin, and sulfamethoxazole-trimethoprim. Unless otherwise indicated, all bacteria were grown in Luria-Bertani (LB) medium. For anaerobic culture, bacteria were grown in a sealed container with AnaeroPack-Anaero carbon dioxide gas generators (Mitsubishi Gas Chemical Co., Inc., Tokyo, Japan). The cell growth was monitored by determining the absorbance at 600 nm. For marker selection and maintenance of plasmids, antibiotics were added to growth media at the following concentrations: 30 $\mu$g/mL chloramphenicol and 50 $\mu$g/mL kanamycin.

**Cloning and mutant constructions.** In-frame deletion mutants of *ackA*, *pta*, *glpT*, *uhpT*, *glpR*, *cpxAR*, and *fis* were constructed by sequence overlap extension PCR according to a strategy described previously (53) using delta1/delta2 and delta3/delta4 primer pairs for each gene, as described in Table S3. The upstream flanking DNA included 450 bp and the first four amino acid codons for *ackA*, *glpT*, and *uhpT*, the first three amino acid codons for *pta*, *glpR*, and *fis*, and six amino acid codons for *cpxR*. The downstream flanking DNA included the last three amino acid codons for *ackA*, and the last two amino acid codons for *pta*, *glpR*, *cpxA*, and *fis*, the last four amino acid codons for *glpT* and *uhpT*, the stop codon, and 450 bp of DNA. These deletion constructs were ligated into BamHI- and Sall-digested temperature-sensitive vector pKO3 (53) and introduced into the UPEC and EHEC strains. Then, sucrose-resistant/chloramphenicol-sensitive colonies were selected at 30°C. We also constructed an *ackA pta* double deletion mutant using primer pairs ackA-delta1/ackApta-delta2 and ackApta-delta3/pta-delta4. To construct the *ackA*, *pta*, and *fis* expression plasmids pTrc99KackA, pTrc99Kpta, and pTrc99Kfis, the *ackA*, *pta*, and *fis* genes were amplified with the primer pairs shown in Table S3. The products were digested with NcoI and BamHI for *ackA* and *fis* and NcoI and BglII for *pta* and ligated into similarly digested pTrc99K plasmid (54). His$_6$-Fis expression plasmid pQE80fis was constructed by ligating the *fis* gene, PCR amplified with the primers pQE80fis-F and pQE80fis-R (see Table S3), into BamHI- and HindIII-digested pQE80L plasmid. All constructs were confirmed by DNA sequencing.

**Drug susceptibility assays.** MIC assays were performed by a serial agar dilution method according to the standard method of the Clinical and Laboratory Standards Institute (CLSI). The MICs were determined as the lowest concentration at which growth was inhibited. The assays were repeated three times for each strain. Bacterial survival after drug exposure was also evaluated. Bacteria were grown in LB medium to the midlogarithmic phase, then $10^8$ cells were transferred into the fresh LB medium containing 25 mg/L fosfomycin. At the indicated time points, aliquots of the cultures were sampled, and the number of viable cells was determined as the CFU by plating serial dilutions on LB agar. Percentage survival was calculated as the value of CFU for cells after incubated with drug relative to the value of CFU for cells prior to incubation with drug as time zero.

**Fosfomycin active transport assays.** Assays to test fosfomycin accumulation in bacterial cells were conducted as previously described (24). Bacteria were grown in 20 mL of LB medium to late-logarithmic phase and resuspended in 1 mL of LB. This suspension was incubated for 60 min at 37°C in the presence

of 2 mg of fosfomycin per mL and then washed three times with hypertonic buffer (10 mM Tris [pH 7.3], 0.5 mM $MgCl_2$, and 150 mM NaCl) to remove the antibiotic. Cells were resuspended in 0.5 mL of distilled water and plated on LB agar to determine the number of CFU/mL. The resuspended bacteria were boiled at 100°C for 3 min to release the fosfomycin. After centrifugation, the antibiotic concentration in the supernatant was determined by a diffusion disc assay. In this assay, sterilized assay discs (13 mm; Whatman, Florham Park, NJ) were saturated with 0.1 mL of the supernatant and deposited onto LB agar plates overlaid with a 1:10 dilution of an overnight culture of *E. coli* MG1655 as a reporter strain. Commercial fosfomycin was used as a standard (FUJIFILM Wako Pure Chemical Industries Corp., Osaka, Japan). Fosfomycin concentration in supernatants was quantified by the diameter (mm) of inhibitory zones on the LB agar culture and represented as $\mu$g per $10^8$ cells.

**RNA extraction and quantitative real-time PCR analyses.** Bacteria were grown to the late-logarithmic growth phase (optical density at 600 nm [$OD_{600}$] ~0.7) in LB medium. Total RNA extraction and cDNA synthesis were performed by using the Monarch Total RNA Miniprep kit (New England Biolabs, Ipswich, MA) and ReverTra Ace qPCR RT Master Mix with gDNA Remover (Toyobo Co. Ltd., Osaka, Japan). Real-time PCR mixtures included 2 ng of cDNA and 160 nM primers in Thunderbird Next SYBR qPCR Mix (Toyobo). Constitutively expressed *rrsA* and *rpoD* genes were used as an internal control. The primers are listed in Table S3.

**cAMP assay.** Intracellular cAMP levels were determined by using a cAMP enzyme immunoassay kit (Cayman Chemical, Ann Arbor, MI). Bacteria were grown in 20 mL of LB medium to late logarithmic phase and harvested. The cell pellets were once washed in phosphate-buffered saline, suspended in 0.5 mL of the enzyme immunoassay buffer supplied in the kit, and then lysed by sonication. The lysate was centrifuged, and the amount of cAMP in the resulting supernatant was quantified according to the manufacturer's protocol. cAMP levels in fraction of the cell lysate were represented as pmol per cells producing 1 $\mu$g of protein.

**Indole assay.** Indole levels in bacterial culture were determined as previously described (55). Bacteria were grown in the LB medium for 24 h, and indole in the culture supernatants was extracted with ethyl acetate. Then, 0.7 mL of Kovac's reagent was added to 0.02 mL of the ethyl acetate fraction containing indole. The indole concentration was calculated by absorbance at 540 nm according to the standard curve using commercial indole (Fujifilm Wako Pure Chemical Industries Corp., Osaka, Japan).

**Promoter assay.** UPEC strains carrying pNNglpT-P, the LacZ reporter plasmid (25), were grown to the late-logarithmic growth phase ($OD_{600}$ ~0.8) in LB medium. $\beta$-Galactosidase activities from *lacZ* expression were determined as Miller's method (56).

**Purification of His6-Fis.** $His_6$-Fis was expressed in *E. coli* Rosetta(DE3). Bacteria carrying the pQE80fis plasmid were grown at 37°C to an $OD_{600}$ of 0.4 in LB medium, 0.5 mM IPTG was added, and culture growth was continued for 18 h at 16°C. Cells were harvested and lysed in EzBactYeastCrusher containing 60 mg/L lysozyme (ATTO, Tokyo, Japan). The lysate was centrifuged, and the resulting supernatant was mixed with Ni-NTA agarose (Qiagen, Valencia, CA). The agarose was washed with 10 mM imidazole twice and 50 mM imidazole once, and then protein was eluted with 500 mM imidazole. The protein concentration was determined using a protein assay (Bio-Rad, Hercules, CA).

**Gel shift assay.** We used a DNA probe containing the 300-bp upstream region of the *glpT* start codon as used in our previous study (24). We also used a DNA fragment from the *Pseudomonas aeruginosa rhlR* gene as a nonspecific control probe. The probe DNA fragments (0.30 pmol) were mixed with the $His_6$-Fis protein in a 10-$\mu$L reaction mixture containing 20 mM Tris (pH 7.5), 50 mM KCl, 1 mM dithiothreitol, and 10% glycerol. After incubation for 20 min at room temperature, samples were separated by electrophoresis on a 5% nondenaturing acrylamide gel (Hi-QRAS Gel N; Gellex International Co. Ltd., Tokyo, Japan) in Tris-borate-EDTA buffer at 4°C. DNA bands in the gel were stained using 20,000-fold-diluted Novel Green Plus (BIO-HELIX Co., Ltd., Keelung City Taiwan) and visualized under UV light at 300 nm.

## SUPPLEMENTAL MATERIAL

Supplemental material is available online only.

**SUPPLEMENTAL FILE 1**, DOC file, 0.1 MB.
**SUPPLEMENTAL FILE 2**, PDF file, 1.5 MB.
**SUPPLEMENTAL FILE 3**, PDF file, 0.7 MB.
**SUPPLEMENTAL FILE 4**, PDF file, 0.2 MB.

## ACKNOWLEDGMENTS

This research was funded by a Japan Society for the Promotion of Science Grant-in-Aid for Scientific Research (B) (grant 22H02864 to H.H. and M.M.), the Japan Agency Research and Development (grants 22fk0108604h0902 and 22wm0225008h0203 to H.T.), and a Health Labor Sciences Research Grant (grant 21KA1004 to H.T.).

H.H. and H.T. designed the research and wrote the manuscript. H.H., A.T., Y.S., S.H., Y.H., and N.O. performed the experiments. H.H., Y.A.M., M.M., and H.T. analyzed the data. All authors contributed to the article and approved the submitted version.

We declare that we have no known competing financial interests or personal relationships that could have appeared to influence the work reported here.

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
