## [Reviewer comments · Microbiology Spectrum]

Microbiology Spectrum

Inactivation of *ackA* and *pta* genes reduces GlpT expression and susceptibility to fosfomycin in *Escherichia coli*

Hidetada Hirakawa, Ayako Takita, Yumika Sato, Suguru Hiramoto, Yusuke Hashimoto, Noriyasu Ohshima, Yoji Minamishima, Masami Murakami, and Haruyoshi Tomita

Corresponding Author(s): Hidetada Hirakawa, Gunma University Graduate School of Medicine

Review Timeline:

Submission Date:	December 9, 2022
Editorial Decision:	February 6, 2023
Revision Received:	April 11, 2023
Editorial Decision:	April 22, 2023
Revision Received:	April 27, 2023
Accepted:	April 29, 2023

Editor: Po-Yu Liu

Reviewer(s): Disclosure of reviewer identity is with reference to reviewer comments included in decision letter(s). The following individuals involved in review of your submission have agreed to reveal their identity: Laichuang Han (Reviewer #1)

Transaction Report:

DOI: <https://doi.org/10.1128/spectrum.05069-22>

February 6, 2023

Dr. Hidetada Hirakawa
Gunma University Graduate School of Medicine
Department of Bacteriology
3-39-22 Showa-machi
Maebashi, Gunma 3718511
Japan

Re: Spectrum05069-22 (Inactivation of ackA and pta genes reduces GlpT expression and susceptibility to fosfomycin in Escherichia coli)

Dear Dr. Hidetada Hirakawa:

Thank you for submitting your manuscript to Microbiology Spectrum. The reviewers have major suggestions for improvement. In particular, some of the assertions and hypotheses made in the manuscript are not adequately supported by the data presented. To strengthen the manuscript, please include more data. The suggestions include but are not limited to information on the original transposon screen and adding a "model" figure to help the reader understand the various regulatory elements involved in fosR. Additionally, a supplementary section could be added for any "data not shown" that the authors would prefer not to include in the manuscript. I kindly request that the suggestions be given careful consideration, so the manuscript will be a valuable contribution to the field.

Link Not Available

Sincerely,

Po-Yu Liu

Journals Department
Reviewer comments:

Reviewer #1 (Comments for the Author):

Fosfomycin is still effective even for the E. coli strains that are resistant to commonly used antimicrobials. In this study, by screening a transposon mutant library of E. coli causing urinary tract infections, the authors found that mutation of the ackA and

pta genes rendered *E. coli* less susceptible to fosfomycin. Although the authors identified this new phenomenon, it is regrettable that they lacked a rigorous and in-depth exploration of the mechanisms behind it.

Reviewer #2 (Comments for the Author):

Manuscript overview

In this manuscript, Hirakawa et al. found that transposon insertions into the *ackA* or *pta* genes lead to a 4-fold increase in MIC for fosfomycin in *E. coli*. They then make these knockouts in CFT073 and in two clinical strains and show that the MICs in the individual knockouts of *ackA* and *pta* and the double knockout of both genes also display this increased resistance to fosfomycin. The knockout phenotype is also present when the cells are grown anaerobically. Complementation studies are done showing that replacement of the genes by expression from a plasmid causes reversion to the original fosfomycin sensitive phenotype. Additional genetic studies indicate that when the transporter *GlpT* is knocked out, knocking out *ackA* or *pta* has no effect on fosfomycin MIC, indicating that the effect of knocking out *ackA* and *pta* requires the presence of *GlpT*. In contrast, another transporter, *UhpT*, the transcriptional regulator *GlpR*, and the two-component response system proteins *CpxAR*, are dispensable for the increase in MIC seen with *ackA* and *pta* knockouts.

In the molecular biology part of the paper, the authors demonstrate that intracellular accumulation of fosfomycin is impaired in the two knockout strains, and hypothesize that, in light of the genetic results with the *GlpT* knockout, *glpT* transcription may be down-regulated in the *ackA* and *pta* knockout strains. In Figure 3, they show that indeed *glpT* transcripts are lower in the *ackA* and *pta* deletion strains, though *murA* mRNA levels (*MurA* is the target of fosfomycin) are unchanged. They also measure *uhpT* transcript levels, and although it appears to me that it is quite a bit lower in the *pta* knockout, the authors conclude that *uhpT* transcription is also unchanged in the mutants.

The last data looks at cAMP and indole levels in the wt and mutant strains. These molecules represent potential regulators of *glpT* expression. Both appear to be unchanged in the mutants, supporting the authors' conclusion that mutants' main mechanism for inducing fosfomycin resistance is through down-regulation of *glpT* by an unknown mechanism.

Overall, I think this work has general relevance, particularly in light of the fact that fosfomycin is being investigated for systemic use against MDR strains (Silver, Cold Spring Harb Perspect Med 2017;7:a025262). In this context it will be important to understand the various ways fosfomycin resistance can arise, and these two genes do not appear to have been linked to *fosR* before. However, the manuscript is a little light on data, and I feel more context is needed to bolster the findings. For example, all the "data not shown" could be included in the main text or in a new supplemental section (which I highly recommend adding). For context, I feel more information is needed about the original transposon screen, and I'd recommend adding a "model" figure. In addition, there are places where the writing should be clarified. I will list my suggestions in order of line numbering below. (Line numbers correspond to those in the merged PDF.)

Specific suggestions:

Line 72: Replace "used" for "using"

Line 74: Rephrase to something like, "there is a shortage of..."

Line 80: "re-attracted" must be following by something, such as "re-attracted attention"

Line 85: I read in this excellent review (Silver, Cold Spring Harb Perspect Med 2017;7:a025262), that *FosC* is actually a version of *FomA*. Please review this finding and edit this line as necessary. Also, I could not find evidence that *fosX* is found on plasmids, only on IS elements. Please review this as well and edit wording as necessary (for example, you could say "mobile elements" in place of "plasmids" if necessary). You may need to include updated references.

Line 86-87: Consider grammar changes to read: "In addition to these exogenous mechanisms..."

Line 93: Add parenthesis around "(required for cAMP production)" for clarity as was done in line 221, and then omit this phrase from line 221 because it will already have been defined here.

Lines 82-103: This is a very long paragraph. Consider breaking it up, perhaps after the sentence ending in line 94.

Lines 82-103: Many different genes are mentioned. Perhaps a real expert in the field of fosfomycin resistance will be familiar with them all, but most readers won't. I feel a diagram of the various mechanisms would be helpful, showing what's up-regulated or down-regulated by what, including small molecule mediators as well as proteins. Perhaps this would be nice as a final figure, also serving as a model for your conclusion that *AckA* and *Pta* regulate *glpT* expression by an as yet unknown mechanism (that could be represented with a question mark.) This final figure could be referenced here in the intro to give the reader a visual of the different mechanisms involved in *fosR*.

Lines 82-103: In reviewing the literature, I found evidence that *PtsI* (Nilsson et al, ANTIMICROBIAL AGENTS AND CHEMOTHERAPY, Sept. 2003, p. 2850-2858; Martin-Gutierrez, Antibiotics 2020, 9(11), 802) and *UhpB* and *UhpC* (Cattoir et al, Front. Microbiol. 11:575031) are also involved in *fosR* but these proteins receive no mention here. They don't have to be included if the authors think this is too much information, but I'm curious as to the reason for omitting some mechanisms while including others.

Lines 105, 109, 118: Replace "the fosfomycin activity" with simply, "fosfomycin activity".

Lines 123-127. I'd really like to know more about the original transposon screen. Is there a reference for the details of this particular screen from your laboratory? If so, please include the reference. If not, I think at least the following details should be included here:

1) What transposon was used? Does it have an outward promoter as part of the sequence to minimize downstream effects?

2) How saturated do you expect this screen of 2200 colonies to be? In other words, is this likely to represent a collection where almost every non-essential gene should be knocked out at least once?

3) You mention that two strains had insertions into *ackA* and *pta*. What other insertions were isolated in this screen? Will there be a future paper about them? Did you also get insertions into the expected genes (*glpT*, *cyaA* etc)? If no other insertions besides the expected ones and your *ackA* and *pta* isolates were observed, that would be a compelling result worth mentioning.

4) I found two other articles that used transposon mutagenesis to look for effects on fosfomycin susceptibility: Turner et al, *J Antimicrob Chemother* 2020; 75: 3144-3151; and Coward et al, *FEMS Microbiology Letters*, 367, 2020, fnaa185. These are both fairly recent studies and seem relevant to your work. Were these genes found in those screens? Glancing through those papers it seems they weren't, although I didn't go through all the data and supplemental materials in detail. Why do you think they weren't found in these or other screens? Do you think those studies should be mentioned here?

Line 136: Should say that the survival rates correspond to x hours post-exposure at x-times the MIC of the parent strain (you fill in the x's).

Line 138: Replace "estimated" with "determined" or "measured".

Line 146: I really think the MICs to these other antibiotics should be shown either in Table 1, as another separate table in the paper, or at the very least in the supplemental material.

Lines 148-170: Paragraph too long. Consider breaking it up, possibly after sentence ending in line 161.

Lines 165-170: Flesh this out a little more for those not so well versed in bacterial genetics. What do you expect if the genes are involved in the same pathway vs. if they act independently of each other?

Line 169: I would suggest softening this statement to something like: "Altogether, we conclude that reduced susceptibility..."

Lines 172-191: Paragraph too long. Consider breaking it up.

Line 179: Consider adding the word "Also," at the beginning of sentence: "Also, deletion of *ackA* and *pta* genes..."

Lines 182-191: Is the point of this section to explain why acetyl-phosphate and/or acetyl-CoA was not measured? I found this section somewhat confusing. Perhaps state up-front (good place to start a new paragraph) that the effects probably aren't due to an accumulation of acetyl-phosphate, which can regulate CpxR, and then state your reasoning.

Line 189: Consider replacing "as well" with "to the same extent" for clarity.

Line 191: Put this data into Figure 3, or as a supplemental figure.

Lines 193: Need to reword the title of this section for clarity. For example, "Deletion of *ackA* and *pta* also results in increased resistance to fosfomycin in an ESBL-producing strain and in an enterohemorrhagic strain of *E. coli*." (You don't really mean these deletions result in susceptibility, right?)

Lines 203-206: It would beef up the paper a bit if you could include these new, unpublished sequences in supplemental material (that of the *ackA-pta* and *glpT* regions of GU2019-E4).

Line 208: This sentence: "The sequences were >96% identical," what does it refer to? Identical to Sakai or to the original CFT073 strain? Or all to each other? Please clarify.

Line 211: Consider adding some kind of closing remark such as, "... multidrug resistant strain and EHEC, providing potential clinical relevance to mutations in these genes."

Line 217: Consider starting the sentence with "The transporters *GlpT* and *UhpT* are required..."

Line 219: Add "expression levels" in place of "expression" for clarity.

Line 221: Can omit "(require for the cAMP synthesis)" because this was stated earlier.

Line 223: Consider adding "also" as in "... *AckA* and *Pta* also contribute to expression..."

Line 225: Change "the source" to "a source".

Line 231: Switch word order to "another uncharacterized regulatory molecule".

Line 239: Make this more clear. For example, "...less sensitive to fosfomycin as compared to the wild-type parent strain to the same extent as was the *ackA* single-deficient strain..."

Line 257: Typo: two "that"s in a row. Consider: "...the possibility that such mutants are generated..."

Line 268: Do you think there is a way to check for clinical relevance of mutations in these genes? Has it been done before? It might be worth pointing the reader in a future direction as a parting remark.

Tables and Figures:

Tables 4 and 5 seem more like they belong as supplemental material. In their place, you could add a table with MICs for the other, FOS-unrelated drugs that were tested as a new table or add that data into Table 1.

Figures 2, 4, and 5 could be combined into one figure with three panels since they all deal with intracellular concentrations of small molecules.

Figure 3 could be beefed up by the addition of *torC* transcript level data, or add this data to supplementary material.

A potential last figure (Figure 4, if you combine figures 2, 4, and 5 into one) could be the "model" figure with all the proteins and small molecule regulators of *fosR*, including your new findings, *AckA* and *Pta*.

I also highly recommend adding a Supplemental Material section that could include, for example:

- 1) Tables 4 and 5
- 2) MIC data for Fos-unrelated drugs (if it's not placed in the main manuscript)
- 3) Sequences of *ackA-pta* and *glpT* regions of GU2019-E4
- 4) The *torC* transcript data if it doesn't go into Figure 3

Staff Comments:

Preparing Revision Guidelines

Please return the manuscript within 60 days; if you cannot complete the modification within this time period, please contact me. If you do not wish to modify the manuscript and prefer to submit it to another journal, please notify me of your decision immediately so that the manuscript may be formally withdrawn from consideration by Microbiology Spectrum.

Fosfomycin is still effective even for the *E. coli* strains that are resistant to commonly used antimicrobials. However, several factors have been reported to reduce the susceptibility of *E. coli* to fosfomycin. In this study, by screening a transposon mutant library of *E. coli* causing urinary tract infections, the authors found that mutation of the *ackA* and *pta* genes rendered *E. coli* less susceptible to fosfomycin. Further investigation revealed that the deletion of *ackA* and *pta* resulted in the down-regulation of the expression of GlpT, a transporter protein of fosfomycin, as the main reason for the resistance of *E. coli* to fosfomycin. The present study reported novel mutations that reduce the action of fosfomycin, although the frequency of mutations in *ackA* and *pta* is low, as the authors pointed out.

Although the authors identified this new phenomenon, they lacked a rigorous and in-depth exploration of the mechanisms behind it. In addition, this work has less data to support some of the assertions and hypotheses made by the authors in this article. From my point of view, this article is of insufficient quality to be published in *Microbiology Spectrum*.

The reasons for my assessment and the issues I hope the author will study in depth are as follows:

1. The deletion of *ackA* and *pta* decreased the expression of GlpT, but the its reason was not well resolved. I think it is important for understanding the resistance of *E. coli* to fosfomycin.

The results of this study indicated that GlpT was the main protein that transports fosfomycin, compared to UhpT. The authors found that *ackA* and *pta* deletion decreased the expression level of GlpT (Fig. 3), resulting in the resistance of *E. coli* to fosfomycin. The levels of cAMP and indole did not vary significantly in different strains, and theoretically *ackA* and *pta* do not affect GlpT expression by regulating their levels. By comparing MICs of *glpR*-deficient strains, the authors concluded that *ackA* and *pta* also do not act through GlpR. Base on these results, the authors discussed and speculated that “metabolic changes may cause decreased *glpT* expression via uncharacterized regulatory molecules.

I can hardly agree with this view. First, it is unconvincing to determine from the changes in MICs alone that GlpR is not involved in the regulation of GlpT by *ackA* and *pta*. The authors preferably determined the transcript levels of *glpR* and its regulated genes in *ackA* and *pta*-deficient strains. More importantly, there are more factors that regulate *glpT* expression. According to the database RegulonDB, the DNA-binding transcriptional dual regulator FNR,

factor for inversion stimulation FIS and Integration host factor IHF can also regulate *glpT* expression, in addition to GlpR and cAMP-regulated CRP, (<http://regulondb.ccg.unam.mx/operon?term=ECK120014760&organism=ECK12&format=jsp&type=operon>). It is worth noting that FNR activates genes involved in anaerobic metabolism and represses genes involved in aerobic metabolism, and also regulates the genes expression involved in the acid resistance, including *ackA*, *pta* and *glpT*. The authors should examine these existing regulatory factors, so as not to mislead other researchers who are interested in the issue.

2. In the section of Introduction, the authors pointed out “We aimed to obtain further insight into the bacterial genes associated with *glpT* and *uhpT* expression that affect the fosfomycin activity” (line 104-105). I do not believe that the authors' strategy of screening fosfomycin-sensitive mutants from the transposon libraries is consistent with the above purpose.

3. There was no clear description of how the transposon library was constructed, nor was there a citation for it.

4. The authors used RT-qPCR to characterize the transcript levels of genes, however, the method of data analysis was not clearly described. The relevant methods are also not clearly described in the cited literature.

5. Some abbreviations was not defined at their first mention, such as MICs (line 125) and the protein MurA(line 83).

6. I think “data not shown” is inappropriate for the testing of other antibiotics (line 146).

Review of: Inactivation of *ackA* and *pta* genes reduces GlpT expression and susceptibility to fosfomycin in *Escherichia coli* by Hirakawa et al.

February 2023

Manuscript overview

In this manuscript, Hirakawa et al. found that transposon insertions into the *ackA* or *pta* genes lead to a 4-fold increase in MIC for fosfomycin in *E. coli*. They then make these knockouts in CFT073 and in two clinical strains and show that the MICs in the individual knockouts of *ackA* and *pta* and the double knockout of both genes also display this increased resistance to fosfomycin. The knockout phenotype is also present when the cells are grown anaerobically. Complementation studies are done showing that replacement of the genes by expression from a plasmid causes reversion to the original fosfomycin sensitive phenotype. Additional genetic studies indicate that when the transporter GlpT is knocked out, knocking out *ackA* or *pta* has no effect on fosfomycin MIC, indicating that the effect of knocking out *ackA* and *pta* requires the presence of GlpT. In contrast, another transporter, UhpT, the transcriptional regulator GlpR, and the two-component response system proteins CpxAR, are dispensable for the increase in MIC seen with *ackA* and *pta* knockouts.

In the molecular biology part of the paper, the authors demonstrate that intracellular accumulation of fosfomycin is impaired in the two knockout strains, and hypothesize that, in light of the genetic results with the GlpT knockout, *glpT* transcription may be down-regulated in the *ackA* and *pta* knockout strains. In Figure 3, they show that indeed *glpT* transcripts are lower in the *ackA* and *pta* deletion strains, though *murA* mRNA levels (MurA is the target of fosfomycin) are unchanged. They also measure *uhpT* transcript levels, and although it appears to me that it is quite a bit lower in the *pta* knockout, the authors conclude that *uhpT* transcription is also unchanged in the mutants.

The last data looks at cAMP and indole levels in the wt and mutant strains. These molecules represent potential regulators of *glpT* expression. Both appear to be unchanged in the mutants, supporting the authors' conclusion that mutants' main mechanism for inducing fosfomycin resistance is through down-regulation of *glpT* by an unknown mechanism.

Overall, I think this work has general relevance, particularly in light of the fact that fosfomycin is being investigated for systemic use against MDR strains (Silver, Cold Spring Harb Perspect Med 2017;7:a025262). In this context it will be important to understand the various ways fosfomycin resistance can arise, and these two genes do not appear to have been linked to *fos*^R before. However, the manuscript is a little light on data, and I feel more context is needed to bolster the findings. For example, all the "data not shown" could be included in the main text or in a new supplemental section (which I highly recommend adding). For context, I feel more information is needed about the original transposon screen, and I'd recommend adding a "model" figure. In addition, there are places where the writing should be clarified. I will list my suggestions in order of line numbering below. (Line numbers correspond to those in the merged PDF.)

Specific suggestions:

Line 72: Replace “used” for “using”

Line 74: Rephrase to something like, “there is a shortage of...”

Line 80: “re-attracted” must be following by something, such as “re-attracted attention”

Line 85: I read in this excellent review (Silver, Cold Spring Harb Perspect Med 2017;7:a025262), that FosC is actually a version of FomA. Please review this finding and edit this line as necessary. Also, I could not find evidence that *fosX* is found on plasmids, only on IS elements. Please review this as well and edit wording as necessary (for example, you could say “mobile elements” in place of “plasmids” if necessary). You may need to include updated references.

Line 86-87: Consider grammar changes to read: “In addition to these exogenous mechanisms...”

Line 93: Add parenthesis around “(required for cAMP production)” for clarity as was done in line 221, and then omit this phrase from line 221 because it will already have been defined here.

Lines 82-103: This is a very long paragraph. Consider breaking it up, perhaps after the sentence ending in line 94.

Lines 82-103: Many different genes are mentioned. Perhaps a real expert in the field of fosfomycin resistance will be familiar with them all, but most readers won't. I feel a diagram of the various mechanisms would be helpful, showing what's up-regulated or down-regulated by what, including small molecule mediators as well as proteins. Perhaps this would be nice as a final figure, also serving as a model for your conclusion that AckA and Pta regulate *glpT* expression by an as yet unknown mechanism (that could be represented with a question mark.) This final figure could be referenced here in the intro to give the reader a visual of the different mechanisms involved in fos^R.

Lines 82-103: In reviewing the literature, I found evidence that PtsI (Nilsson et al, ANTIMICROBIAL AGENTS AND CHEMOTHERAPY, Sept. 2003, p. 2850–2858; Martin-Gutierrez, *Antibiotics* 2020, 9(11), 802) and UhpB and UhpC (Cattoir et al, Front. Microbiol. 11:575031) are also involved in fos^R but these proteins receive no mention here. They don't have to be included if the authors think this is too much information, but I'm curious as to the reason for omitting some mechanisms while including others.

Lines 105, 109, 118: Replace “the fosfomycin activity” with simply, “fosfomycin activity”.

Lines 123-127. I'd really like to know more about the original transposon screen. Is there a reference for the details of this particular screen from your laboratory? If so, please include the reference. If not, I think at least the following details should be included here:

- 1) What transposon was used? Does it have an outward promoter as part of the sequence to minimize downstream effects?
- 2) How saturated do expect this screen of 2200 colonies to be? In other words, is this likely to represent a collection where almost every non-essential gene should be knocked out at least once?
- 3) You mention that two strains had insertions into *ackA* and *pta*. What other insertions were isolated in this screen? Will there be a future paper about them? Did you also get insertions into the expected genes (*glpT*, *cyaA* etc)? If no other insertions besides the expected ones and your *ackA* and *pta* isolates were observed, that would be a compelling result worth mentioning.

- 4) I found two other articles that used transposon mutagenesis to look for effects on fosfomycin susceptibility: Turner et al, J Antimicrob Chemother 2020; 75: 3144–3151; and Coward et al, FEMS Microbiology Letters, 367, 2020, fnaa185. These are both fairly recent studies and seem relevant to your work. Were these genes found in those screens? Glancing through those papers it seems they weren't, although I didn't go through all the data and supplemental materials in detail. Why do you think they weren't found in these or other screens? Do you think those studies should be mentioned here?

Line 136: Should say that the survival rates correspond to x hours post-exposure at x-times the MIC of the parent strain (you fill in the x's).

Line 138: Replace "estimated" with "determined" or "measured".

Line 146: I really think the MICs to these other antibiotics should be shown either in Table 1, as another separate table in the paper, or at the very least in the supplemental material.

Lines 148-170: Paragraph too long. Consider breaking it up, possibly after sentence ending in line 161.

Lines 165-170: Flesh this out a little more for those not so well versed in bacterial genetics. What do you expect if the genes are involved in the same pathway vs. if they act independently of each other?

Line 169: I would suggest softening this statement to something like: "Altogether, we conclude that reduced susceptibility..."

Lines 172-191: Paragraph too long. Consider breaking it up.

Line 179: Consider adding the word "Also," at the beginning of sentence: "Also, deletion of *ackA* and *pta* genes..."

Lines 182-191: Is the point of this section to explain why acetyl-phosphate and/or acetyl-CoA was not measured? I found this section somewhat confusing. Perhaps state up-front (good place to start a new paragraph) that the effects probably aren't due to an accumulation of acetyl-phosphate, which can regulate CpxR, and then state your reasoning.

Line 189: Consider replacing "as well" with "to the same extent" for clarity.

Line 191: Put this data into Figure 3, or as a supplemental figure.

Lines 193: Need to reword the title of this section for clarity. For example, "Deletion of *ackA* and *pta* also results in increased resistance to fosfomycin in an ESBL-producing strain and in an enterohemorrhagic strain of *E. coli*." (You don't really mean these deletions result in susceptibility, right?)

Lines 203-206: It would beef up the paper a bit if you could include these new, unpublished sequences in supplemental material (that of the *ackA-pta* and *glpT* regions of GU2019-E4).

Line 208: This sentence: "The sequences were >96% identical," what does it refer to? Identical to Sakai or to the original CFT073 strain? Or all to each other? Please clarify.

Line 211: Consider adding some kind of closing remark such as, "... multidrug resistant strain and EHEC, providing potential clinical relevance to mutations in these genes."

Line 217: Consider starting the sentence with “The transporters GlpT and UhpT are required...”

Line 219: Add “expression levels” in place of “expression” for clarity.

Line 221: Can omit “(require for the cAMP synthesis)” because this was stated earlier.

Line 223: Consider adding “also” as in “... AckA and Pta also contribute to expression...”

Line 225: Change “the source” to “a source”.

Line 231: Switch word order to “another uncharacterized regulatory molecule”.

Line 239: Make this more clear. For example, “...less sensitive to fosfomycin as compared to the wild-type parent strain to the same extent as was the *ackA* single-deficient strain...”

Line 257: Typo: two “that”s in a row. Consider: “...the possibility that such mutants are generated...”

Line 268: Do you think there is a way to check for clinical relevance of mutations in these genes? Has it been done before? It might be worth pointing the reader in a future direction as a parting remark.

Tables and Figures:

Tables 4 and 5 seem more like they belong as supplemental material. In their place, you could add a table with MICs for the other, FOS-unrelated drugs that were tested as a new table or add that data into Table 1.

Figures 2, 4, and 5 could be combined into one figure with three panels since they all deal with intracellular concentrations of small molecules.

Figure 3 could be beefed up by the addition of *torC* transcript level data, or add this data to supplementary material.

A potential last figure (Figure 4, if you combine figures 2, 4, and 5 into one) could be the “model” figure with all the proteins and small molecule regulators of *fos*^R, including your new findings, *AckA* and *Pta*.

I also highly recommend adding a Supplemental Material section that could include, for example:

- 1) Tables 4 and 5
- 2) MIC data for Fos-unrelated drugs (if it’s not placed in the main manuscript)
- 3) Sequences of *ackA-pta* and *glpT* regions of GU2019-E4
- 4) The *torC* transcript data if it doesn’t go into Figure 3

Reviewer1

Fosfomycin is still effective even for the *E. coli* strains that are resistant to commonly used antimicrobials. However, several factors have been reported to reduce the susceptibility of *E. coli* to fosfomycin. In this study, by screening a transposon mutant library of *E. coli* causing urinary tract infections, the authors found that mutation of the *ackA* and *pta* genes rendered *E. coli* less susceptible to fosfomycin. Further investigation revealed that the deletion of *ackA* and *pta* resulted in the down-regulation of the expression of GlpT, a transporter protein of fosfomycin, as the main reason for the resistance of *E. coli* to fosfomycin. The present study reported novel mutations that reduce the action of fosfomycin, although the frequency of mutations in *ackA* and *pta* is low, as the authors pointed out. Although the authors identified this new phenomenon, they lacked a rigorous and in-depth exploration of the mechanisms behind it. In addition, this work has less data to support some of the assertions and hypotheses made by the authors in this article. From my point of view, this article is of insufficient quality to be published in *Microbiology Spectrum*.

The reasons for my assessment and the issues I hope the author will study in depth are as follows:

Response: Thank you for your constructive comments and suggestions. We deepened this study according to your suggestions. We found that mutations in *ackA* and *pta* reduced expression of the nucleotide-associated protein Fis. We further demonstrated that Fis promotes the expression of *glpT* by a series of experiments including MIC assay, quantitative PCR, promoter test, and gel shift assay. Eventually, we conclude that elevated fosfomycin resistance by mutations in *ackA* and *pta* is associated with reduction of Fis expression. We believe that this raises the quality of this work to the level of that which can be published in *Microbiology Spectrum*.

1. The deletion of *ackA* and *pta* decreased the expression of GlpT, but the its reason was not well resolved. I think it is important for understanding the resistance of *E. coli* to fosfomycin. The results of this study indicated that GlpT was the main protein that transports fosfomycin, compared to UhpT. The authors found that *ackA* and *pta* deletion decreased the expression level of GlpT (Fig. 3), resulting in the resistance of *E. coli* to fosfomycin. The levels of cAMP and indole did not vary significantly in different strains, and theoretically *ackA* and *pta* do not affect GlpT expression by regulating their levels. By comparing MICs of *glpR*-deficient strains, the authors concluded that *ackA* and *pta* also do not act through GlpR. Base on these results, the authors discussed and speculated that “metabolic changes may cause decreased *glpT* expression *via* uncharacterized regulatory molecules.

I can hardly agree with this view. First, it is unconvincing to determine from the changes in MICs alone that GlpR is not involved in the regulation of GlpT by *ackA* and *pta*. The authors preferably determined the transcript levels of *glpR* and its regulated genes in *ackA* and *pta*-deficient strains. More importantly, there are more factors that regulate *glpT* expression. According to the database RegulonDB, the DNA-binding transcriptional dual regulator FNR, factor for inversion stimulation FIS and Integration host factor IHF can also regulate *glpT* expression, in addition to GlpR and cAMP-regulated CRP, (<http://regulondb.ccg.unam.mx/operon?term=ECK120014760&organism=ECK12&format=json&type=operon>). It is worth noting that FNR activates genes involved in anaerobic metabolism and represses genes involved in aerobic metabolism, and also regulates the genes expression involved in the acid resistance, including *ackA*, *pta* and *glpT*. The authors should examine these existing regulatory factors, so as not to mislead other researchers who are interested in the issue.

Response: Using the RegulonDB as a reference, we compared transcription levels between parent and mutant strains by qPCR for several genes presumed to contribute to *glpT* expression in addition to *glpR*. We found that the transcription level of the *fis* gene was moderately decreased in the *ackA* and *pta* mutant strains (Approximately 50% decrease) (Figure 3B). We found that expression of the *fis* gene on a plasmid in the *ackA* and *pta* mutant strains reduced the MIC of fosfomycin to the same level as in the parent strain (Table 1). Furthermore, no reduction in MIC was observed when the *fis* gene was exogenously expressed on the *glpT* mutant (Table 1). A decrease in promoter activity of *glpT* was also observed when the *fis* gene was deleted (Figure 4A). Purified Fis protein was also found to bind to the upstream region of *glpT* (Figure 4B). These results indicate that Fis activates the expression of *glpT*, but that the *ackA* and *pta* mutations reduce the level of Fis, which in turn reduces the expression of *glpT*, resulting in reduced fosfomycin activity. We also mentioned in our discussion that FNR is involved in the regulation of *ackA* and *pta* as well as acid resistance genes.

2. In the section of Introduction, the authors pointed out “We aimed to obtain further insight into the bacterial genes associated with *glpT* and *uhpT* expression that affect the fosfomycin activity”(line 104-105). I do not believe that the authors' strategy of screening fosfomycin-sensitive mutants from the transposon libraries is consistent with the above purpose.

Response: Our purpose is to identify novel genes that affect fosfomycin activity in UPEC. Fosfomycin activity is dependent on the GlpT and UhpT transporters. Since GlpT and UhpT are secondary transporters, their expression is thought to be regulated by various

transcriptional regulators, including known regulators. On the other hand, fosfomycin inhibits the function of MurA, which gene is considered to be house-keeping since *murA* is an essential gene. For this reason, we hypothesize that the screened set of genes contributing to fosfomycin resistance includes a significant number of genes that affect the expression of GlpT and UhpT. As pointed out, the original text may mislead the readers, then we simply changed it to “We aimed to obtain further insight into the bacterial genes that affect fosfomycin activity”.

3. There was no clear description of how the transposon library was constructed, nor was there a citation for it.

Response: We used the Epicentre EZ-Tn5<R6Kγori/KAN-2>Tnp Transposome Kit for our previous study (Kurabayashi et al., Infect Immun., 2016). Originally, we performed transposon mutagenesis with this kit to find out genes that are involved in biofilm formation. In the process, we obtained 2200 clones with transposon inserts. We used this clone sets for this study. We explained it in Introduction and the top of Result.

4. The authors used RT-qPCR to characterize the transcript levels of genes, however, the method of data analysis was not clearly described. The relevant methods are also not clearly described in the cited literature.

Response: We conducted the RNA extraction and reverse transcription with same reagents as those used in Hirakawa et al., Sci Rep. 2020. However, the reference did not clearly describe the method for real-time PCR. We now provided the information for this experiment in the text.

5. Some abbreviations was not defined at their first mention, such as MICs (line 125) and the protein MurA(line 83).

Response: As suggested, we defined MICs and MurA as minimum inhibitory concentrations and UDP-N-acetylglucosamine-3- enolpyruvyltransferase, respectively.

6. I think “data not shown” is inappropriate for the testing of other antibiotics (line 146).

Response: As suggested, the MICs were presented in Supplementary table.

Reviewer2

Overall, I think this work has general relevance, particularly in light of the fact that fosfomycin is being investigated for systemic use against MDR strains (Silver, Cold Spring Harb Perspect Med 2017;7:a025262). In this context it will be important to understand the various ways fosfomycin resistance can arise, and these two genes do not appear to have been linked to fos before. However, the manuscript is a little light on data, and I feel more context is needed to bolster the findings. For example, all the “data not shown” could be included in the main text or in a new supplemental section (which I highly recommend adding). For context, I feel more information is needed about the original transposon screen, and I’d recommend adding a “model” figure. In addition, there are places where the writing should be clarified. I will list my suggestions in order of line numbering below. (Line numbers correspond to those in the merged PDF.)

Response: Thank you for your constructive comments and suggestions. Accordingly, we revised our manuscript providing additional information and data including the fis deletion and overexpression analyses and binding of the Fis protein to *glpT* promoter region.

Specific suggestions:

Line 72: Replace “used” for “using”

Response: Replaced.

Line 74: Rephrase to something like, “there is a shortage of...”

Response: We rephrased it to “there is a shortage of new alternative agents”.

Line 80: “re-attracted” must be following by something, such as “re-attracted attention”

Response: We added “attention” after re-attracted.

Line 85: I read in this excellent review (Silver, Cold Spring Harb Perspect Med 2017;7:a025262), that FosC is actually a version of FomA. Please review this finding and edit this line as necessary. Also, I could not find evidence that *fosX* is found on plasmids, only on IS elements. Please review this as well and edit wording as necessary (for example, you could say “mobile elements” in place of “plasmids” if necessary). You may need to include updated references.

Response: Thank you for letting us know. We now update our knowledge. We accordingly modified line85 with citing the review. In addition, we deleted “plasmids” for the *fosX* statement.

Line 86-87: Consider grammar changes to read: “In addition to these exogenous mechanisms...”

Response: Changed.

Line 93: Add parenthesis around “(required for cAMP production)” for clarity as was done in line 221, and then omit this phrase from line 221 because it will already have been defined here.

Response: As suggested, we added parenthesis and deleted (Required for the cAMP synthesis) from the sentence in line 221 sentence.

Lines 82-103: This is a very long paragraph. Consider breaking it up, perhaps after the sentence ending in line 94.

Response: As suggested, it was split into two paragraphs.

Lines 82-103: Many different genes are mentioned. Perhaps a real expert in the field of fosfomycin resistance will be familiar with them all, but most readers won't. I feel a diagram of the various mechanisms would be helpful, showing what's up-regulated or down-regulated by what, including small molecule mediators as well as proteins. Perhaps this would be nice as a final figure, also serving as a model for your conclusion that AckA and Pta regulate *g/pT* expression by an as yet unknown mechanism (that could be represented with a question mark.) This final figure could be referenced here in the intro to give the reader a visual of the different mechanisms involved in fos_R.

Response: As suggested, we provided additional figure that describes the regulatory complex of GlpT/UhpT involved in the fosfomycin activity including our conclusion (Figure 5).

Lines 82-103: In reviewing the literature, I found evidence that PtsI (Nilsson et al, *ANTIMICROBIAL AGENTS AND CHEMOTHERAPY*, Sept. 2003, p. 2850–2858; Martin-Gutierrez, *Antibiotics* 2020, *9*(11), 802) and UhpB and UhpC (Cattoir et al, *Front. Microbiol.* 11:575031) are also involved in fos_R but these proteins receive no mention here. They don't have to be included if the authors think this is too much information, but I'm curious as to the reason for omitting some mechanisms while including others.

Response: We included these literatures related to PtsI, UhpB and UhpC. Thank you for catching them.

Lines 105, 109, 118: Replace “the fosfomycin activity” with simply, “fosfomycin activity”.

Response: Changed.

Lines 123-127. I’d really like to know more about the original transposon screen. Is there a reference for the details of this particular screen from your laboratory? If so, please include the reference. If not, I think at least the following details should be included here:

1) What transposon was used? Does it have an outward promoter as part of the sequence to minimize downstream effects?

2) How saturated do you expect this screen of 2200 colonies to be? In other words, is this likely to represent a collection where almost every non-essential gene should be knocked out at least once?

Response: We used the Epicentre EZ-Tn5<R6Kγori/KAN-2>Tnp Transposome Kit. We used Tn5, and it has a promoter sequence at the upstream of the kanamycin resistance gene, which does not exclude a possibility that a transposon insertion affects the transcription of downstream genes. For a transposon insert into the *ackA* gene, expression of the *pta* gene may be affected. However, we constructed the “clean deletion mutants” for *ackA* and *pta* genes. We also confirmed that susceptibility to fosfomycin was complemented by exogenous gene expression. 2) We think that this screen of 2200 colonies is not saturated. Originally, we performed transposon mutagenesis with this kit to find out genes that are involved in biofilm formation. In the process, we obtained 2200 clones with transposon inserts. We used this clone sets for this study (We explained it in Introduction and the start of Result). Therefore, our screen has a limitation. However, our aim is not to conduct comprehensive screening to find out genes that is responsible for fosfomycin activity. In addition to searching for new genes that contribute to fosfomycin activity, we are focusing on elucidating the detailed mechanisms of resistance.

3) You mention that two strains had insertions into *ackA* and *pta*. What other insertions were isolated in this screen? Will there be a future paper about them? Did you also get insertions into the expected genes (*glpT*, *cyaA* etc)? If no other insertions besides the expected ones and your *ackA* and *pta* isolates were observed, that would be a compelling result worth mentioning.

Response: We got clones that have Tn-insertions into *cyaA* and *glpT* in addition to *ackA* and *pta*. We mentioned this result in the Result section.

4) I found two other articles that used transposon mutagenesis to look for effects on fosfomycin susceptibility: Turner et al, J Antimicrob Chemother 2020; 75: 3144–3151; and

Coward et al, FEMS Microbiology Letters, 367, 2020, fnaa185. These are both fairly recent studies and seem relevant to your work. Were these genes found in those screens? Glancing through those papers it seems they weren't, although I didn't go through all the data and supplemental materials in detail. Why do you think they weren't found in these or other screens? Do you think those studies should be mentioned here?

Response: Thank you for letting us know. We carefully read these papers including the data presented in the supplemental materials. Both of *ackA* and *pta* were not found. While those studies screened in the presence of fosfomycin, our study identified fosfomycin-resistant strains by measuring MICs individually for clones obtained in the absence of fosfomycin. Compared to the former method, our method can identify modest resistant strains with a 4-fold increase in MIC from the parent strain, although the number of transposon-insert mutants that can be screened in a single experiment is limited compared to the former method. We believe that this provides us with mutant strains (*ackA* and *pta* mutants) not obtained in previous studies. We mentioned those studies in Discussion.

Line 136: Should say that the survival rates correspond to x hours post-exposure at x-times the MIC of the parent strain (you fill in the x's).

Response: We added a sentence beginning with “The survival rates correspond to 1 to 3 hours post-exposure at 6.25-times the MIC of the parent strain were measured”.

Line 138: Replace “estimated” with “determined” or “measured”.

Response: We replaced “estimated” with “determined”.

Line 146: I really think the MICs to these other antibiotics should be shown either in Table 1, as another separate table in the paper, or at the very least in the supplemental material.

Response: As suggested, the MICs to drugs except fosfomycin were presented in Supplementary table (Table S1).

Lines 148-170: Paragraph too long. Consider breaking it up, possibly after sentence ending in line 161.

Response: As suggested, it was split into two sentences.

Lines 165-170: Flesh this out a little more for those not so well versed in bacterial genetics. What do you expect if the genes are involved in the same pathway vs. if they act independently of each other?

Response: Fosfomycin is taken up separately via two different transporters, GlpT and UhpT. We tested whether the increased resistance to fosfomycin caused by deletion of *ackA* and *pta* is associated with the GlpT or UhpT transport pathway. Based on the qPCR results, we hypothesized that no further increase in resistance to fosfomycin is caused by genetic deletion of *ackA* and *pta* in the absence of *glpT*, since we believe that the increased resistance to fosfomycin is caused by decreased GlpT expression. We added this explanation to the text.

Line 169: I would suggest softening this statement to something like: “Altogether, we conclude that reduced susceptibility...”

Response: We accordingly modified it.

Lines 172-191: Paragraph too long. Consider breaking it up.

Response: We extended our qPCR analysis with additional genes. Then, we reconstructed this section.

Line 179: Consider adding the word “Also,” at the beginning of sentence: “Also, deletion of *ackA* and *pta* genes...”

Response: As suggested, we added “Also”.

Lines 182-191: Is the point of this section to explain why acetyl-phosphate and/or acetyl-CoA was not measured? I found this section somewhat confusing. Perhaps state up-front (good place to start a new paragraph) that the effects probably aren’t due to an accumulation of acetyl-phosphate, which can regulate CpxR, and then state your reasoning.

Response: Yes, it is. As you pointed out, this section confuses readers. We deleted this section.

Line 189: Consider replacing “as well” with “to the same extent” for clarity.

Response: Corrected.

Line 191: Put this data into Figure 3, or as a supplemental figure.

Response: We included into the figure (Figure 3B).

Lines 193: Need to reword the title of this section for clarity. For example, “Deletion of *ackA* and *pta* also results in increased resistance to fosfomycin in an ESBL-producing strain and in an enterohemorrhagic strain of *E. coli*.” (You don’t really mean these deletions result in

susceptibility, right?)

Response: Thank you for letting us know. The title was fixed.

Lines 203-206: It would beef up the paper a bit if you could include these new, unpublished sequences in supplemental material (that of the *ackA-pta* and *glpT* regions of GU2019-E4).

Response: We provided the sequence into supplemental material (Figure S1-3).

Line 208: This sentence: "The sequences were >96% identical," what does it refer to? Identical to Sakai or to the original CFT073 strain? Or all to each other? Please clarify.

Response: It means that the sequences were >96% identical among the three strains. We modified that sentence.

Line 211: Consider adding some kind of closing remark such as, "... multidrug resistant strain and EHEC, providing potential clinical relevance to mutations in these genes."

Response: As suggested, we added this phrase.

Line 217: Consider starting the sentence with "The transporters GlpT and UhpT are required..."

Response: As suggested, we added "The transporters".

Line 219: Add "expression levels" in place of "expression" for clarity.

Response: We accordingly added it.

Line 221: Can omit "(require for the cAMP synthesis)" because this was stated earlier.

Response: Deleted.

Line 223: Consider adding "also" as in "... AckA and Pta also contribute to expression..."

Response: We accordingly added it.

Line 225: Change "the source" to "a source".

Response: Corrected.

Line 231: Switch word order to "another uncharacterized regulatory molecule".

Response: We found that Fis is involved in reduction of *glpT* expression by *ackA* and *pta* mutations. Therefore, we deleted the original sentence.

Line 239: Make this more clear. For example, "...less sensitive to fosfomycin as compared to

the wildtype parent strain to the same extent as was the *ackA* single-deficient strain..."

Response: We accordingly changed it.

Line 257: Typo: two "that"'s in a row. Consider: "...the possibility that such mutants are generated..."

Response: Corrected.

Line 268: Do you think there is a way to check for clinical relevance of mutations in these genes? Has it been done before? It might be worth pointing the reader in a future direction as a parting remark.

Response: We agree that that is a good approach for the future direction. No clinically isolated fosfomycin-resistant strains with mutations in *ackA* and *pta* have been reported. There are no reports on the association between *ackA/pta* and antimicrobial resistance, not only to fosfomycin, and we assume that no one has paid attention to *ackA/pta* in drug resistance studies. As NGS analysis is becoming more accessible, it is possible to find out the fosfomycin-resistant strains, with *ackA* and *pta* SNPs, that have been isolated so far and those that will be isolated in the future if we look for mutations in *ackA* and *pta* in those resistant strains with a particular attention. We added it to the end of Discussion.

Tables and Figures:

Tables 4 and 5 seem more like they belong as supplemental material. In their place, you could add a table with MICs for the other, FOS-unrelated drugs that were tested as a new table or add that data into Table 1.

Response: As suggested, we moved Tables 4 and 5 to supplemental material (Table S2 and S3), and MIC data for other drugs was provided as a supplementary table (Table S1).

Figures 2, 4, and 5 could be combined into one figure with three panels since they all deal with intracellular concentrations of small molecules.

Response: As suggested, we combined these figures (Figure 2A-C).

Figure 3 could be beefed up by the addition of *torC* transcript level data, or add this data to supplementary material.

Response: We included the *torC* data in the qPCR figure (Figure 3B).

A potential last figure (Figure 4, if you combine figures 2, 4, and 5 into one) could be the "model" figure with all the proteins and small molecule regulators of *fosR*, including your new

findings, AckA and Pta.

Response: As suggested, we provided the model figure (Figure 5).

I also highly recommend adding a Supplemental Material section that could include, for example:

- 1) Tables 4 and 5
- 2) MIC data for Fos-unrelated drugs (if it's not placed in the main manuscript)
- 3) Sequences of ackA-pta and glpT regions of GU2019-E4
- 4) The torC transcript data if it doesn't go into Figure 3

Response: As suggested, in addition to tables 4 and 5 contexts, MIC data and sequences were provided in supplemental data (Table S1-3). The *torC* data was included in the qPCR figure (Figure 3B).

April 22, 2023

Dr. Hidetada Hirakawa
Gunma University Graduate School of Medicine
Department of Bacteriology
3-39-22 Showa-machi
Maebashi, Gunma 3718511
Japan

Re: Spectrum05069-22R1 (Inactivation of ackA and pta genes reduces GlpT expression and susceptibility to fosfomycin in Escherichia coli)

Dear Dr. Hidetada Hirakawa:

Link Not Available

Sincerely,

Po-Yu Liu

Journals Department
Reviewer comments:

Reviewer #2 (Comments for the Author):

I am reviewing the resubmission of this paper after having reviewed the original manuscript. In this resubmission, the authors have addressed many of my concerns and also added experimental information about the involvement of the Fis protein in regulating glpT expression. In particular, I was pleased to see further explanation of the transposon screen, the MICs of other antibiotics in Supplemental Table S1 as well as the addition of the "data not shown" into supplemental material, and the inclusion of a model figure (Figure 5), which adds much clarity to the complex regulatory mechanisms at work in this system. I still have some suggestions (below) many of which are geared toward the clarity of writing. In particular, the Discussion needs some reworking. Also, I feel Figure 5, which is a great addition, needs to be made into one panel and the regulatory network described with additional color coding.

I have the following comments (Line numbers based on clean, resubmitted manuscript):

Line 45: Add comma: "nucleoid-associated protein, Fis." Also correct spelling to "nucleoid" here and throughout document.

Line 45-47: Awkward sentence structure making it less clear what your conclusion is. Consider changing to something more direct like: "We found that mutations in *ackA* and *pta* also caused a decrease in *fis* expression. Thus, we interpret the decrease in *glpT* expression in *ackA* and *pta* knockout strains to be due to a decrease in *Fis* levels in these mutants. "

Lines 92-93: Consider wording change to "...*GlpT*, *UhpT* and *MurA* on the chromosome alter susceptibility to the drug. In particular, functional deletion..."

Lines 96-113. This is a long paragraph with a lot of information in it, which may be hard to digest for a reader not familiar with the field. It would help to walk the reader through it a little, and refer to your new model figure (Fig. 5). Some suggestions:

1. First sentence, consider: "Fosfomycin sensitivity is also attenuated by decreased expression of *GlpT* and *UhpT*, which can occur by several mechanisms (Figure 5)."

2. Line 102-103, reduced expression of *uhpT* presumably comes first (not consequent to) *fosR*. Consider reversing the order of this sentence to something like, "...were found to result in reduced expression of *uhpT*, with consequent resistance to fosfomycin."

3. Start new paragraph at line 103 sentence, "In addition 104 to the positive regulators..." Could also reference Figure 5 at the end of this sentence, i.e., "...act as repressors for the *glpT* and *uhpT* genes (Figure 5)."

4. I would add the abbreviation: "... trimethylamine-N-oxide (TMAO)..." since it is labeled that way in the figure.

Line 122: Correct to "acetate"

Line 141-142: Were these the only two clones you found with substantial resistance besides the obvious ones? Was your cut-off for "moderate resistance" 4-fold? Again, if these were the only two in your collection that were 4-fold or over, that would be nice to state here. You could say, "In addition to these clones, we found only two other clones..." Also, I feel you should report an estimate for how much coverage you expect your library of 2200 clones to represent, i.e., 50% (or whatever the number is) of expected non-essential genes. Maybe this is reported in the original paper - if so, refer to that number here or elsewhere. (You could consider adding the information about the transposon screen into a small paragraph in the Methods section if it seems like too much to put into the results section. Then you could describe in detail how you screened the transposon library and what cut-offs were etc.)

Line 143: Consider starting a new paragraph at "To verify this result..."

Line 149-152: Consider keeping this to one sentence: "We also compared the survival of the mutant strains with that of the parent strain at 1 to 3 hours post-exposure at 6.25-times the MIC of the parent strain."

Line 182: Split into multiple sentences for clarity: "Based on the qPCR results, we hypothesized that the increased resistance to fosfomycin seen in the *ackA* and *pta* mutants is caused mostly by decreased *glpT* expression. If so, then we would expect to see no further increase in resistance to fosfomycin upon deletion of *ackA* and *pta* in the absence of *glpT*, while the absence *uhpT* should not impact the effects of these mutations on fosfomycin sensitivity."

Lines 188-189: For clarity, use fold-increases here: "...deletion of *ackA* and *pta* from the *uhpT* mutant resulted in a further 4-fold increase in fosfomycin MICs (Table 1)."

Line 204: Change to "activates".

Line 222: Need to point out this lower than the parent strain: "reduced the MIC to 2 mg/L, 2-fold lower than the parent strain (Table 1)."

Lines 225-226 : Would it be fair to add the following?: "These observations suggest that the increased susceptibility to fosfomycin due to *fis* expression is dependent on *glpT*, and that *AckA* and *Pta* act upstream of *Fis* in the regulation of *glpT*."

Line 234: I definitely see a shift at 1 pmol with the *rhIR* control construct, but I agree that it could be non-specific. I would still say the shift can be seen at 1 pmol (not at 2 pmol) with the control as compared to 0.25 with *glpT*.

Lines 235-238: Perhaps soften these statements since the EMSA results are not super-convincing: "These results suggest that *Fis* binds to the upstream region of *glpT*, increasing its promoter activity. Altogether, we conclude *ackA* and *pta* mutations cause a decrease in...."

Line 249-250: Add your Supplemental reference here: "...the sequence of the *ackA-pta* operon for GU2019-E4 was determined 249 because the complete genome sequence for this strain has not been available (Figure S1)." Then delete this sentence: "We found orthologous *ackA* and *pta* genes in the GU2019-E4 chromosome."

Line 254-255: Similarly, add Supplemental ref here: "...for GU2019-E4 were also determined and compared with those for CFT073 and EHEC O157:H7 Sakai strains (Figure S2, S3)." Then drop ref after next sentence: "...three strains." Also note extra space in line 254.

Line 264: This first sentence is awkward. You could say something like, "It is known that chromosomal mutations can affect fosfomycin sensitivity."

Line 269: Note extra space.

Line 270-274. Needs to be reworded for clarity. How about: "The current study shows that *AckA* and *Pta* also contribute to expression of *glpT* and sensitivity to fosfomycin because deletion of the *ackA* and *pta* genes reduced expression of *glpT*, but not *uhpT*, and decreased sensitivity to the drug. Therefore, the *ackA* and *pta* genes may be a source of fosfomycin resistance."

Lines 275-284: While I'm glad you acknowledge these studies here, it's not necessary to say which are better or worse. Better to keep it short and let the reader explore for themselves. Here is a possible re-wording of this section (please use your own judgment/wording here, this is just an example of how to reduce this section to only the key points):

"Recently, two comprehensive studies combining transposon mutagenesis with next-generation sequencing (NGS) analysis identified genes contributing to fosfomycin activity [35] [36]. Interestingly, *ackA* and *pta* were not identified in these studies. One

difference between our study and these is that in the other studies, mutants were selected to survive in the presence of fosfomycin while in our study mutants were obtained in the absence of fosfomycin. These differences could account for the differences in results obtained."

Line 284: Start a new paragraph with "Expression..."

Line 286: Need to be more clear about what data you are considering to make this statement. For example, you could say: "According to results from our knockout strain fosfomycin sensitivity assays, small molecule (cAMP and indole) measurements, and qPCR gene expression analyses, the decreased expression of glpT due to ackA and pta gene inactivation likely does not involve these regulators."

Line 290: End sentence at "...glpT expression." Next sentence, "In a series of experiments including MIC assays of fis deletion mutants, fis expression qPCR analyses, promoter expression assays, and gel shift assays, we showed that Fis increases glpT expression."

Line 298: Change to "other regulators" or "additional regulators" rather than "another regulators".

Line 306: Extra "t"

Lines 311-315: This paragraph seems out of place here. I would put it instead in between the paragraphs in lines 298-299.

Lines 314-315: Consider changing to: "...free CoA pool [42], which may cause decreased fis expression."

Line 320: Extra space

Line 327: Use "disadvantageous" or "a disadvantage" in place of "disadvantage".

Line 329: Change "its" to "their".

Lines 331-337: These last sentences need to be more clear. Consider something like:

"As more NGS data becomes available, it will be possible check if fosfomycin-resistant strains that have been isolated so far and those that will be isolated in the future have mutations in ackA and pta. We believe that this study will aid us to more precisely predict fosfomycin resistance and thereby help to keep fosfomycin treatment effective."

Line 376: Did you do your MIC measurements more than once for each strain? If so, please state the minimum number of MIC measurements taken to establish each strain's MIC. I hope this is at least 3 times for each strain, as MICs are known to drift by 2-fold here and there.

Lines 676, 679, 682: I would use "MICs" rather than "MIC" in the titles of the tables, but use "MIC" rather than "MICs" in the "Strain" line (line below 676, line below 679, and line below 682).

Line 682: You can say, "Table 3. Fosfomycin MICs of other clinical E. coli strains" rather than just "other strains".

Line 713: Extra "_"

Line 716: Is this supposed to say *, $P \leq 0.05$?

Line 723: Need to define what "P" and "TMAO" and "G6P" stand for in the figure.

Figure 4A: Label Y-axis as " β -galactosidase activity" not "activities"

Figure 5: While I think this figure is a great addition to the paper, it doesn't quite unify all the data as it is now. Would it be possible to also color code the second level regulators, such as TorT/TorS, CpxA, UhpC/B, and all the small molecules, or does this get too confusing? Also, I'd rather see Pta and AckA put into the top part of figure 5 and have only one panel. If you need more room, you could put the regulators all around glpT and uhpT in a circle, rather than having them all above. For the Pta and AckA effect, you could direct an arrow toward Fis, which would then direct toward glpT (all with appropriate colors), and also an arrow with a question mark directly to glpT to represent the unknown additional regulator. You could highlight your contribution in some way (bold, circled etc) to show what you've added to this picture.

Staff Comments:

Preparing Revision Guidelines

Please return the manuscript within 60 days; if you cannot complete the modification within this time period, please contact me. If you do not wish to modify the manuscript and prefer to submit it to another journal, please notify me of your decision

immediately so that the manuscript may be formally withdrawn from consideration by Microbiology Spectrum.

Review of “Inactivation of *ackA* and *pta* genes reduces GlpT expression and susceptibility to fosfomycin in *Escherichia coli*” Resubmission

4/16/2023

I am reviewing the resubmission of this paper after having reviewed the original manuscript. In this resubmission, the authors have addressed many of my concerns and also added experimental information about the involvement of the Fis protein in regulating *glpT* expression. In particular, I was pleased to see further explanation of the transposon screen, the MICs of other antibiotics in Supplemental Table S1 as well as the addition of the “data not shown” into supplemental material, and the inclusion of a model figure (Figure 5), which adds much clarity to the complex regulatory mechanisms at work in this system. I still have some suggestions (below) many of which are geared toward the clarity of writing. In particular, the Discussion needs some reworking. Also, I feel Figure 5, which is a great addition, needs to be made into one panel and the regulatory network described with additional color coding.

I have the following comments (Line numbers based on clean, resubmitted manuscript):

Line 45: Add comma: “nucleoid-associated protein, Fis.” Also correct spelling to “nucleoid” here and throughout document.

Line 45-47: Awkward sentence structure making it less clear what your conclusion is. Consider changing to something more direct like: “We found that mutations in *ackA* and *pta* also caused a decrease in *fis* expression. Thus, we interpret the decrease in *glpT* expression in *ackA* and *pta* knockout strains to be due to a decrease in Fis levels in these mutants. ”

Lines 92-93: Consider wording change to “...GlpT, UhpT and MurA on the chromosome alter susceptibility to the drug. In particular, functional deletion...”

Lines 96-113. This is a long paragraph with a lot of information in it, which may be hard to digest for a reader not familiar with the field. It would help to walk the reader through it a little, and refer to your new model figure (Fig. 5). Some suggestions:

1. First sentence, consider: “Fosfomycin sensitivity is also attenuated by decreased expression of GlpT and UhpT, which can occur by several mechanisms (Figure 5).”
2. Line 102-103, reduced expression of *uhpT* presumably comes first (not consequent to) *fosR*. Consider reversing the order of this sentence to something like, “...were found to result in reduced expression of *uhpT*, with consequent resistance to fosfomycin.”
3. Start new paragraph at line 103 sentence, “In addition 104 to the positive regulators...” Could also reference Figure 5 at the end of this sentence, i.e., “...act as repressors for the *glpT* and *uhpT* genes (Figure 5).”
4. I would add the abbreviation: “... trimethylamine-N-oxide (TMAO)...” since it is labeled that way in the figure.

Line 122: Correct to “acetate”

Line 141-142: Were these the only two clones you found with substantial resistance besides the obvious ones? Was your cut-off for “moderate resistance” 4-fold? Again, if these were the only two in your collection that were 4-fold or over, that would be nice to state here. You could say, “In addition to these clones, we found only two other clones...” Also, I feel you should report an estimate for how much coverage you expect your library of 2200 clones to represent, i.e., 50% (or whatever the number is) of expected non-essential genes. Maybe this is reported in the original paper – if so, refer to that number here or elsewhere. (You could consider adding the information about the transposon screen into a small paragraph in the Methods section if it seems like too much to put into the results section. Then you could describe in detail how you screened the transposon library and what cut-offs were etc.)

Line 143: Consider starting a new paragraph at “To verify this result...”

Line 149-152: Consider keeping this to one sentence: “We also compared the survival of the mutant strains with that of the parent strain at 1 to 3 hours post-exposure at 6.25-times the MIC of the parent strain.”

Line 182: Split into multiple sentences for clarity: “Based on the qPCR results, we hypothesized that the increased resistance to fosfomycin seen in the *ackA* and *pta* mutants is caused mostly by decreased *glpT* expression. If so, then we would expect to see no further increase in resistance to fosfomycin upon deletion of *ackA* and *pta* in the absence of *glpT*, while the absence *uhpT* should not impact the effects of these mutations on fosfomycin sensitivity.”

Lines 188-189: For clarity, use fold-increases here: “...deletion of *ackA* and *pta* from the *uhpT* mutant resulted in a further 4-fold increase in fosfomycin MICs (Table 1).”

Line 204: Change to “activates”.

Line 222: Need to point out this lower than the parent strain: “reduced the MIC to 2 mg/L, 2-fold lower than the parent strain (Table 1).”

Lines 225-226 : Would it be fair to add the following?: “These observations suggest that the increased susceptibility to fosfomycin due to *fis* expression is dependent on *glpT*, and that AckA and Pta act upstream of Fis in the regulation of *glpT*.”

Line 234: I definitely see a shift at 1 pmol with the rhIR control construct, but I agree that it could be non-specific. I would still say the shift can be seen at 1 pmol (not at 2 pmol) with the control as compared to 0.25 with *glpT*.

Lines 235-238: Perhaps soften these statements since the EMSA results are not super-convincing: “These results suggest that Fis binds to the upstream region of *glpT*, increasing its promoter activity. Altogether, we conclude *ackA* and *pta* mutations cause a decrease in....”

Line 249-250: Add your Supplemental reference here: “...the sequence of the *ackA-pta* operon for GU2019-E4 was determined 249 because the complete genome sequence for this strain has not been available (Figure S1).” Then delete this sentence: “We found orthologous *ackA* and *pta* genes in the GU2019-E4 chromosome.”

Line 254-255: Similarly, add Supplemental ref here: "...for GU2019-E4 were also determined and compared with those for CFT073 and EHEC O157:H7 Sakai strains (Figure S2, S3)." Then drop ref after next sentence: "...three strains." Also note extra space in line 254.

Line 264: This first sentence is awkward. You could say something like, "It is known that chromosomal mutations can affect fosfomycin sensitivity."

Line 269: Note extra space.

Line 270-274. Needs to be reworded for clarity. How about: "The current study shows that AckA and Pta also contribute to expression of *glpT* and sensitivity to fosfomycin because deletion of the *ackA* and *pta* genes reduced expression of *glpT*, but not *uhpT*, and decreased sensitivity to the drug. Therefore, the *ackA* and *pta* genes may be a source of fosfomycin resistance."

Lines 275-284: While I'm glad you acknowledge these studies here, it's not necessary to say which are better or worse. Better to keep it short and let the reader explore for themselves. Here is a possible rewording of this section (please use your own judgment/wording here, this is just an example of how to reduce this section to only the key points):

"Recently, two comprehensive studies combining transposon mutagenesis with next-generation sequencing (NGS) analysis identified genes contributing to fosfomycin activity [35] [36]. Interestingly, *ackA* and *pta* were not identified in these studies. One difference between our study and these is that in the other studies, mutants were selected to survive in the presence of fosfomycin while in our study mutants were obtained in the absence of fosfomycin. These differences could account for the differences in results obtained."

Line 284: Start a new paragraph with "Expression..."

Line 286: Need to be more clear about what data you are considering to make this statement. For example, you could say: "According to results from our knockout strain fosfomycin sensitivity assays, small molecule (cAMP and indole) measurements, and qPCR gene expression analyses, the decreased expression of *glpT* due to *ackA* and *pta* gene inactivation likely does not involve these regulators."

Line 290: End sentence at "...*glpT* expression." Next sentence, "In a series of experiments including MIC assays of *fis* deletion mutants, *fis* expression qPCR analyses, promoter expression assays, and gel shift assays, we showed that Fis increases *glpT* expression."

Line 298: Change to "other regulators" or "additional regulators" rather than "another regulators".

Line 306: Extra "t"

Lines 311-315: This paragraph seems out of place here. I would put it instead in between the paragraphs in lines 298-299.

Lines 314-315: Consider changing to: "...free CoA pool [42], which may cause decreased *fis* expression."

Line 320: Extra space

Line 327: Use "disadvantageous" or "a disadvantage" in place of "disadvantage".

Line 329: Change "its" to "their".

Lines 331-337: These last sentences need to be more clear. Consider something like:

“As more NGS data becomes available, it will be possible check if fosfomycin-resistant strains that have been isolated so far and those that will be isolated in the future have mutations in *ackA* and *pta*. We believe that this study will aid us to more precisely predict fosfomycin resistance and thereby help to keep fosfomycin treatment effective.”

Line 376: Did you do your MIC measurements more than once for each strain? If so, please state the minimum number of MIC measurements taken to establish each strain’s MIC. I hope this is at least 3 times for each strain, as MICs are known to drift by 2-fold here and there.

Lines 676, 679, 682: I would use “MICs” rather than “MIC” in the titles of the tables, but use “MIC” rather than “MICs” in the “Strain” line (line below 676, line below 679, and line below 682).

Line 682: You can say, “Table 3. Fosfomycin MICs of other clinical *E. coli* strains” rather than just “other strains”.

Line 713: Extra “_”

Line 716: Is this supposed to say *, $P \leq 0.05$?

Line 723: Need to define what “P” and “TMAO” and “G6P” stand for in the figure.

Figure 4A: Label Y-axis as “ β -galactosidase activity” not “activities”

Figure 5: While I think this figure is a great addition to the paper, it doesn’t quite unify all the data as it is now. Would it be possible to also color code the second level regulators, such as TorT/TorS, CpxA, UhpC/B, and all the small molecules, or does this get too confusing? Also, I’d rather see Pta and AckA put into the top part of figure 5 and have only one panel. If you need more room, you could put the regulators all around *glpT* and *uhpT* in a circle, rather than having them all above. For the Pta and AckA effect, you could direct an arrow toward Fis, which would then direct toward *glpT* (all with appropriate colors), and also an arrow with a question mark directly to *glpT* to represent the unknown additional regulator. You could highlight your contribution in some way (bold, circled etc) to show what you’ve added to this picture.

I am reviewing the resubmission of this paper after having reviewed the original manuscript. In this resubmission, the authors have addressed many of my concerns and also added experimental information about the involvement of the Fis protein in regulating *glpT* expression. In particular, I was pleased to see further explanation of the transposon screen, the MICs of other antibiotics in Supplemental Table S1 as well as the addition of the “data not shown” into supplemental material, and the inclusion of a model figure (Figure 5), which adds much clarity to the complex regulatory mechanisms at work in this system. I still have some suggestions (below) many of which are geared toward the clarity of writing. In particular, the Discussion needs some reworking. Also, I feel Figure 5, which is a great addition, needs to be made into one panel and the regulatory network described with additional color coding. I have the following comments (Line numbers based on clean, resubmitted manuscript):

Response: Again, thank you so much for your valuable and constructive comments. Accordingly, we revised our manuscript. We added more information for Tn-screening and improved Figure5 (Now, we renamed Figure1). Our point-by-point responses are described below.

Line 45: Add comma: “nucleoid-associated protein, Fis.” Also correct spelling to “nucleoid” here and throughout document.

Response: Corrected. Thank you for finding our misspellings.

Line 45-47: Awkward sentence structure making it less clear what your conclusion is. Consider changing to something more direct like: “We found that mutations in *ackA* and *pta* also caused a decrease in *fis* expression. Thus, we interpret the decrease in *glpT* expression in *ackA* and *pta* knockout strains to be due to a decrease in Fis levels in these mutants.”

Response: We changed it, accordingly.

Lines 92-93: Consider wording change to “...GlpT, UhpT and MurA on the chromosome alter susceptibility to the drug. In particular, functional deletion...”

Response: We changed it, accordingly.

Lines 96-113. This is a long paragraph with a lot of information in it, which may be hard to digest for a reader not familiar with the field. It would help to walk the reader through it a little, and refer to your new model figure (Fig. 5). Some suggestions:

1. First sentence, consider: “Fosfomycin sensitivity is also attenuated by decreased expression of GlpT and UhpT, which can occur by several mechanisms (Figure 5).”

Response: Accordingly, we changed it. We renamed Figure 5 as Figure1 because the Figure is

now the first one presented.

2. Line 102-103, reduced expression of *uhpT* presumably comes first (not consequent to) *fosR*. Consider reversing the order of this sentence to something like, "...were found to result in reduced expression of *uhpT*, with consequent resistance to fosfomycin."

Response: Accordingly, changed.

3. Start new paragraph at line 103 sentence, "In addition 104 to the positive regulators..." Could also reference Figure 5 at the end of this sentence, i.e., "...act as repressors for the *glpT* and *uhpT* genes (Figure 5)."

Response: Accordingly, changed.

4. I would add the abbreviation: "... trimethylamine-N-oxide (TMAO)..." since it is labeled that way in the figure.

Response: Added.

Line 122: Correct to "acetate"

Response: Corrected. Thank you for catching.

Line 141-142: Were these the only two clones you found with substantial resistance besides the obvious ones? Was your cut-off for "moderate resistance" 4-fold? Again, if these were the only two in your collection that were 4-fold or over, that would be nice to state here. You could say, "In addition to these clones, we found only two other clones..." Also, I feel you should report an estimate for how much coverage you expect your library of 2200 clones to represent, i.e., 50% (or whatever the number is) of expected non-essential genes. Maybe this is reported in the original paper – if so, refer to that number here or elsewhere. (You could consider adding the information about the transposon screen into a small paragraph in the Methods section if it seems like too much to put into the results section. Then you could describe in detail how you screened the transposon library and what cut-offs were etc.)

Response: Yes, they were. Our cut-off was 4-fold. The CFT073 genome contains 5,400 genes. The number of essential genes for culture in LB agar is estimated to be ~460 (doi: 10.1016/j.micres.2022.127202). Our library of 2200 clones represents ~44% of non-essential genes. However, it is hard to precisely describe how much coverage. We included this information related to cut-off into the Result section.

Line 143: Consider starting a new paragraph at "To verify this result..."

Response: Accordingly, we started the new paragraph.

Line 149-152: Consider keeping this to one sentence: “We also compared the survival of the mutant strains with that of the parent strain at 1 to 3 hours post-exposure at 6.25-times the MIC of the parent strain.”

Response: We accordingly reconstructed the sentence.

Line 182: Split into multiple sentences for clarity: “Based on the qPCR results, we hypothesized that the increased resistance to fosfomycin seen in the *ackA* and *pta* mutants is caused mostly by decreased *glpT* expression. If so, then we would expect to see no further increase in resistance to fosfomycin upon deletion of *ackA* and *pta* in the absence of *glpT*, while the absence *uhpT* should not impact the effects of these mutations on fosfomycin sensitivity.”

Response: Accordingly, changed.

Lines 188-189: For clarity, use fold-increases here: “...deletion of *ackA* and *pta* from the *uhpT* mutant resulted in a further 4-fold increase in fosfomycin MICs (Table 1).”

Response: Accordingly, changed.

Line 204: Change to “activates”.

Response: Corrected.

Line 222: Need to point out this lower than the parent strain: “reduced the MIC to 2 mg/L, 2-fold lower than the parent strain (Table 1).”

Response: Accordingly, changed.

Lines 225-226 : Would it be fair to add the following?: “These observations suggest that the increased susceptibility to fosfomycin due to *fis* expression is dependent on *glpT*, and that AckA and Pta act upstream of Fis in the regulation of *glpT*.”

Response: We added it.

Line 234: I definitely see a shift at 1 pmol with the rhIR control construct, but I agree that it could be non-specific. I would still say the shift can be seen at 1 pmol (not at 2 pmol) with the control as compared to 0.25 with *glpT*.

Response: We replace the original statement into “The shift could be seen at 1 pmol with the rhIR probe as compared to 0.25 pmol with the *glpT* probe”.

Lines 235-238: Perhaps soften these statements since the EMSA results are not super-convincing: “These results suggest that Fis binds to the upstream region of *glpT*, increasing its promoter activity. Altogether, we conclude *ackA* and *pta* mutations cause a decrease in...”

Response: As suggested, we changed it, accordingly.

Line 249-250: Add your Supplemental reference here: “...the sequence of the *ackA-pta* operon for GU2019-E4 was determined 249 because the complete genome sequence for this strain has not been available (Figure S1).” Then delete this sentence: “We found orthologous *ackA* and *pta* genes in the GU2019-E4 chromosome.”

Response: We added it.

Line 254-255: Similarly, add Supplemental ref here: “...for GU2019-E4 were also determined and compared with those for CFT073 and EHEC O157:H7 Sakai strains (Figure S2, S3).” Then drop ref after next sentence: “...three strains.” Also note extra space in line 254.

Response: We modified them.

Line 264: This first sentence is awkward. You could say something like, “It is known that chromosomal mutations can affect fosfomycin sensitivity.”

Response: Accordingly, we modified it.

Line 269: Note extra space.

Response: Deleted.

Line 270-274. Needs to be reworded for clarity. How about: “The current study shows that AckA and Pta also contribute to expression of *glpT* and sensitivity to fosfomycin because deletion of the *ackA* and *pta* genes reduced expression of *glpT*, but not *uhpT*, and decreased sensitivity to the drug. Therefore, the *ackA* and *pta* genes may be a source of fosfomycin resistance.”

Response: Accordingly, we changed it.

Lines 275-284: While I’m glad you acknowledge these studies here, it’s not necessary to say which are better or worse. Better to keep it short and let the reader explore for themselves. Here is a possible rewording of this section (please use your own judgment/wording here, this is just an example of how to reduce this section to only the key points):

“Recently, two comprehensive studies combining transposon mutagenesis with

next-generation sequencing (NGS) analysis identified genes contributing to fosfomycin activity [35] [36]. Interestingly, *ackA* and *pta* were not identified in these studies. One difference between our study and these is that in the other studies, mutants were selected to survive in the presence of fosfomycin while in our study mutants were obtained in the absence of fosfomycin. These differences could account for the differences in results obtained.”

Response: Thank you for your advice. Accordingly, we changed the sentence as follows. Interestingly, *ackA* and *pta* were not identified in these studies. One difference between our study and these is that in the other studies, mutants capable of surviving in the presence of fosfomycin were selected while in our study, mutants were obtained in the absence of fosfomycin. These differences could account for the differences in results obtained.

Line 284: Start a new paragraph with “Expression...”

Response: Accordingly, we did it.

Line 286: Need to be more clear about what data you are considering to make this statement. For example, you could say: “According to results from our knockout strain fosfomycin sensitivity assays, small molecule (cAMP and indole) measurements, and qPCR gene expression analyses, the decreased expression of *glpT* due to *ackA* and *pta* gene inactivation likely does not involve these regulators.”

Response: We changed the sentence to According to results from fosfomycin sensitivity assays, small molecule (cAMP and indole) measurements, and qPCR gene expression analyses with our gene knockout strains, the decreased expression of *glpT* due to *ackA* and *pta* gene inactivation likely does not involve these regulators.

Line 290: End sentence at “...*glpT* expression.” Next sentence, “In a series of experiments including MIC assays of *fis* deletion mutants, *fis* expression qPCR analyses, promoter expression assays, and gel shift assays, we showed that Fis increases *glpT* expression.”

Response: We changed it.

Line 298: Change to “other regulators” or “additional regulators” rather than “another regulators”.

Response: We changed another regulators to additional regulators.

Line 306: Extra “t”

Response: Deleted. Thank you.

Lines 311-315: This paragraph seems out of place here. I would put it instead in between the paragraphs in lines 298-299.

Response: The paragraph was moved.

Lines 314-315: Consider changing to: "...free CoA pool [42], which may cause decreased fis expression."

Response: Changed.

Line 320: Extra space

Response: We deleted the space.

Line 327: Use "disadvantageous" or "a disadvantage" in place of "disadvantage".

Response: We changed it to disadvantageous.

Line 329: Change "its" to "their".

Response: Changed.

Lines 331-337: These last sentences need to be more clear. Consider something like:
"As more NGS data becomes available, it will be possible check if fosfomycin-resistant strains that have been isolated so far and those that will be isolated in the future have mutations in *ackA* and *pta*. We believe that this study will aid us to more precisely predict fosfomycin resistance and thereby help to keep fosfomycin treatment effective."

Response: Accordingly changed. Thank you for suggestions.

Line 376: Did you do your MIC measurements more than once for each strain? If so, please state the minimum number of MIC measurements taken to establish each strain's MIC. I hope this is at least 3 times for each strain, as MICs are known to drift by 2-fold here and there.

Response: As you guess, we repeated MIC measurements 3 times for each strain. We stated it in the Method section. At least we observed that MICs in the *ackA* and *pta* mutants were always 4-fold higher than those in the parent strain.

Lines 676, 679, 682: I would use "MICs" rather than "MIC" in the titles of the tables, but use "MIC" rather than "MICs" in the "Strain" line (line below 676, line below 679, and line below 682).

Response: Changed.

Line 682: You can say, “Table 3. Fosfomycin MICs of other clinical *E. coli* strains” rather than just “other strains”.

Response: Changed.

Line 713: Extra “_”

Response: Deleted. Thank you.

Line 716: Is this supposed to say *, $P < 0.05$?

Response: Yes. We stated it as *, $P < 0.05$ relative to the value for CFT073.

Line 723: Need to define what “P” and “TMAO” and “G6P” stand for in the figure.

Response: We defined it as phosphate group, Trimethylamine-N-oxide and Glucose-6-phosphate.

Figure 4A: Label Y-axis as “ β -galactosidase activity” not “activities”

Response: Corrected.

Figure 5: While I think this figure is a great addition to the paper, it doesn't quite unify all the data as it is now. Would it be possible to also color code the second level regulators, such as TorT/TorS, CpxA, UhpC/B, and all the small molecules, or does this get too confusing? Also, I'd rather see Pta and AckA put into the top part of figure 5 and have only one panel. If you need more room, you could put the regulators all around glpT and uhpT in a circle, rather than having them all above. For the Pta and AckA effect, you could direct an arrow toward Fis, which would then direct toward glpT (all with appropriate colors), and also an arrow with a question mark directly to glpT to represent the unknown additional regulator. You could highlight your contribution in some way (bold, circled etc) to show what you've added to this picture.

Response: Accordingly, we modified the figure and its legend..

April 29, 2023

Dr. Hidetada Hirakawa
Gunma University Graduate School of Medicine
Department of Bacteriology
3-39-22 Showa-machi
Maebashi, Gunma 3718511
Japan

Re: Spectrum05069-22R2 (Inactivation of ackA and pta genes reduces GlpT expression and susceptibility to fosfomycin in Escherichia coli)

Dear Dr. Hidetada Hirakawa:

Your manuscript has been accepted, and I am forwarding it to the ASM Journals Department for publication. You will be notified when your proofs are ready to be viewed.

Sincerely,

Po-Yu Liu
Editor, Microbiology Spectrum
